# Multi-scale classification decodes the complexity of the human E3 ligome

Arghya Dutta [1,2,3,5], Alberto Cristiani [1,2,5], Siddhanta V. Nikte [1,2,5], Jonathan Eisert [1,2], Yves Matthess [1], Borna Markusic [1,4], Cosmin Tudose [1], Chiara Becht [1], Ronay Cetin[1], Varun Jayeshkumar Shah[1], Thorsten Mosler [1], Koraljka Husnjak[1], Ivan Dikic [1,2,4], Manuel Kaulich [1] & Ramachandra M. Bhaskara [1,2,4] ✉

E3 ubiquitin ligases are vital enzymes that define the ubiquitin code in cells. Beyond promoting protein degradation to maintain cellular health, they also mediate non-degradative processes like DNA repair, signaling, and immunity. Despite their therapeutic potential, a comprehensive framework for understanding the relationships among diverse E3 ligases is lacking. Here, we classify the "human E3 ligome"−an extensive set of catalytic human E3s−by integrating multi-layered data, including protein sequences, domain architectures, 3D structures, functions, and expression patterns. Our classification is based on a metric-learning paradigm and uses a weakly supervised hierarchical framework to capture authentic relationships across E3 families and subfamilies. It extends the categorization of E3s into RING, HECT, and RBR classes, including non-canonical mechanisms, successfully explains their functional segregation, distinguishes between multi-subunit complexes and standalone enzymes, and maps E3s to substrates and potential drug interactions. Our analysis provides a global view of E3 biology, opening strategies for drugging E3-substrate networks, including drug repurposing and designing specific E3 handles.

Cells constantly modulate their proteomes in response to physiological and environmental changes. The timely removal and turnover of cellular proteins is integral to protein homeostasis[1]. In eukaryotes, individual proteins, complexes, and large assemblies are degraded via either autophagy or the ubiquitin-proteasome system (UPS)[2]. In mammalian cells, ~80% of the cellular proteome is degraded through the UPS[1]. In this pathway, the designated protein cargo is tagged with ubiquitin (Ub) molecules through a series of enzymatic reactions, marking them for degradation by the proteasome[3]. Following the action of E1 and E2 enzymes, the E3 ligase brings both the E2–ubiquitin complex and the substrate protein in proximity, allowing the transfer of Ub from the E2 enzyme to a lysine residue on the target protein[4,5]. This process is often repeated (poly-ubiquitination), resulting in

substrates with distinct types of Ub-chains. In UPS, for instance, K48-linked Ub-chains are recognized by Ub-binding domains (UBDs) on 19S proteasomal particles, initiating the degradation of substrates[1]. In autophagy, ubiquitination often serves as a necessary condition for identifying substrates, conferring specificity[6]. Cargo components, damaged organelles, and intracellular pathogens targeted for degradation are often ubiquitinated. Further, autophagy receptors are enriched in UBDs to recognize modified cargo components[7] or themselves strongly ubiquitinated to trigger aggregation of protein assemblies in the cytosol and organellar membranes[8,9], thus enhancing autophagic flux.

E3 ubiquitin ligases confer substrate specificity for ubiquitination. They recognize distinct targets, operate in diverse cellular locations,

[1]Institute of Biochemistry II, Faculty of Medicine, Goethe University, Theodor-Stern-Kai 7, Frankfurt am Main, Germany. [2]Buchmann Institute for Molecular Life Sciences, Goethe University, Max-von-Laue Str. 15, Frankfurt am Main, Germany. [3]Department of Physics, SRM University–AP, Amaravati, Andhra Pradesh, India. [4]IMPRS on Cellular Biophysics, Max-von-Laue Str. 3, Frankfurt am Main, Germany. [5]These authors contributed equally: Arghya Dutta, Alberto Cristiani, Siddhanta V. Nikte. ✉e-mail: Bhaskara@med.uni-frankfurt.de

and exert spatial control of protein turnover[10,11]. In addition to controlling homeostatic processes, E3 ligases regulate immunity and inflammation pathways[12,13]. Given their tissue-specific expressions and association with developmental and metabolic syndromes, including cancer progression, E3 ligases have emerged as promising candidates, particularly for drugging previously undruggable targets[14]. In stark contrast to E1 (~10) and E2 enzymes (~50), a substantial number of E3 ligases (~600) have been recognized in humans[15,16]. This count of putative E3s stems from various investigations: Li et al.[17] identified ~617 potential human E3-encoding genes by conducting a genome-wide search to detect RING (Really Interesting New Gene) finger catalytic domains using hidden Markov models. Subsequently, Deshaies and Joazeiro[18] characterized ~300 RING and U-box E3 ligases, while Medvar et al.[19] documented ~377 E3 ligases, with a primary focus on confirmed catalytic activity. Despite these efforts, many human E3 ligases have been only partially characterized. A significant fraction remains unexplored and hypothetical or unknown[20]. To date, those studied exhibit extensive heterogeneity in their sequence, domain composition, 3D structure, subcellular localization, and tissue expression, establishing them as one of the most diverse classes of enzymes. Furthermore, several E3 ligases function as multi-subunit complexes with varied substrate specificities modulated by specific receptors, adaptors, and scaffold proteins[21]. The extensive variety and large numbers of E3 ubiquitin ligases create a bottleneck for pattern recognition and large-scale study. Therefore, detailed characterization and analysis of the human E3 ligome—the complete set of E3 ubiquitin ligases encoded by the human genome—is essential for a comprehensive understanding.

The current classification of the E3 ligases—based on the ubiquitin-transfer mechanism—categorizes them into three main classes: RING (Really Interesting New Gene), HECT (Homologous to the E6AP Carboxyl Terminus), and RBR (RING-Between-RING) classes[15]. This classification drastically oversimplifies the mechanistic diversity of E3 ligases, compels the grouping of enzymes with hybrid characteristics, and fails to accommodate emerging information on recent and atypical ligases, limiting its overall utility[18]. A multi-scale classification of the human E3 ligome offers a unique solution to tackle the complexity and remarkable diversity inherent in these enzymes at various scales. This organized approach can provide more accurate and functional groupings, crucial for a nuanced understanding of different E3 ligase families. Further, emerging patterns detected help trace evolutionary relationships more effectively, revealing conserved elements and adaptive changes that are not evident. Furthermore, mapping essential information such as functional diversity, substrate-specificities, and druggability onto the classification provides a global view, guiding specific and directed investigations to fill in the missing information.

Here, we systematically catalog all E3 ubiquitin ligases to build a comprehensive and manually curated human E3 ligome. We then encode the relationships between high-confidence E3 ligases using multiple distance measures at various granular layers spanning the molecular- and the systems-level organization. By amalgamating selected distance measures from multiple layers into an optimized emergent distance metric, we group all human E3 ligases into distinct families and subfamilies. Our classification delineates features and patterns specific to E3 ligase families, providing insights into their organization. By combining CRISPR-Cas9 dropout screens and proteomic analysis with functional enrichment analysis, we identify essential E3s differentially regulated under stress conditions. We demonstrate the utility of this unbiased classification by mapping the existing state of knowledge on E3 ligase domain architecture, 3D structure, function, substrate networks, and small molecule interactions to gain generic and family-specific insights. The multiscale classification framework developed here embodies canonical and atypical E3 mechanisms largely reflecting the ubiquitin code, offering a comprehensive roadmap to navigate the vast landscape of E3 ligase biology, laying the groundwork for future therapeutic applications.

## Results

### Assembly of the human E3 ligome

To comprehensively identify all E3 ligases in the human genome, we conducted a census using datasets from previously published studies and public repositories. By visualizing their overlaps, we found that all existing datasets were largely inconsistent (Fig. 1a and Supplementary Fig. 1a). Most strikingly, only 99 proteins were consistently categorized as human E3 ligases from all eight datasets. The low overlap in these datasets reflects the diverse approaches and often variable definitions used to collate E3 systems (Supplementary Table 1). We resolved these conflicts by explicitly defining the catalytic components of E3 systems, i.e., polypeptide sequences containing one or more catalytic domains ($C = \{d_c\}$, see "Methods"). Using this objective criterion ($\{X_i \in \bigcup_{n=1}^{8} | \exists d_i \in C\}$; Supplementary Table 2) facilitated proper annotation and targeted analysis of E3s. We found that 462 polypeptide sequences, across all datasets ($\bigcup_{n=1}^{8} A_n = 1448$), contain at least one catalytic domain constituting the curated E3 ligome (Fig. 1b and Supplementary Fig. 1b).

To substantiate our curation process, we defined a consensus score for each protein based on its presence in various source datasets (Fig. 1c). We found that the HECT and RBR classes of E3 ligases showed high agreement across datasets (confidence score ≥ 0.6; orange and purple bars). The RING class (green bars) had a broad distribution of consensus scores indicative of annotation challenges. However, the most significant discrepancy among the datasets (confidence score ≤ 0.25) was due to misannotated proteins. E1, E2, and other non-catalytic components of E3 systems, such as receptors, scaffolds, and adaptor proteins, were often merged with E3 ligases, accounting for an additional 298 proteins, leaving 514 unclassified and 174 with no domain annotations (Fig. 1b). Furthermore, several proteins obtained from UniProt and BioGRID using keyword-based searches (Supplementary Fig. 1c) were not included by others, have low consensus scores, and remain unclassified and unannotated, excluding 688 proteins from the curated E3 ligome (Fig. 1c, black bars). Our approach thus minimized false positives and true negatives, includes high-confidence catalytically active E3s and accounts for pseudo-E3 ligases and other E3s with untested catalytic activities, providing a detailed assessment of the completeness of the human E3 ligome (see Supplementary Note 1).

To quantify the diversity of the human E3 ligome, we mapped the sequence, structure, and functional features of individual E3s corresponding to well-known E3 classes (RING, HECT, and RBR). We found that the length distribution of the E3s is broad, ranging from 100 to 5000 residues (mean size = 635 residues; Fig. 1d). The average fractional coverage of E3s annotated with unique domains is 37%, 42%, and 53% for RING, HECT, and RBR classes, respectively (Fig. 1e). Furthermore, on average, the RING, HECT, and RBR domains span 23%, 31%, and 39% of their total lengths, respectively (Fig. 1f). By mapping information from the Protein Data Bank (PDB), we found 2259 distinct structures representing RING, HECT, and RBR-containing proteins (2001+168+90), providing partial structural information for 51% (208+21+8 unique UniProt records) of the E3 ligome (Fig. 1g). Analysis of AlphaFold models revealed that for most E3s, the coverage of structured domains is high, and the amount of intrinsic disorder is generally low (pLDDT ≤ 50 covering only ≤ 10% E3 length; Supplementary Fig. 1d). We quantified the functional diversity of the E3 ligome by retrieving the unique Gene Ontology (GO) annotations corresponding to Biological Processes (BP), Cellular Component (CC), and Molecular Function (MF). We annotated 96–100% of the E3s with unique GO terms (Fig. 1h). The number of distinct GO terms captured the diversity of functional assignments attributed to the three E3 classes.

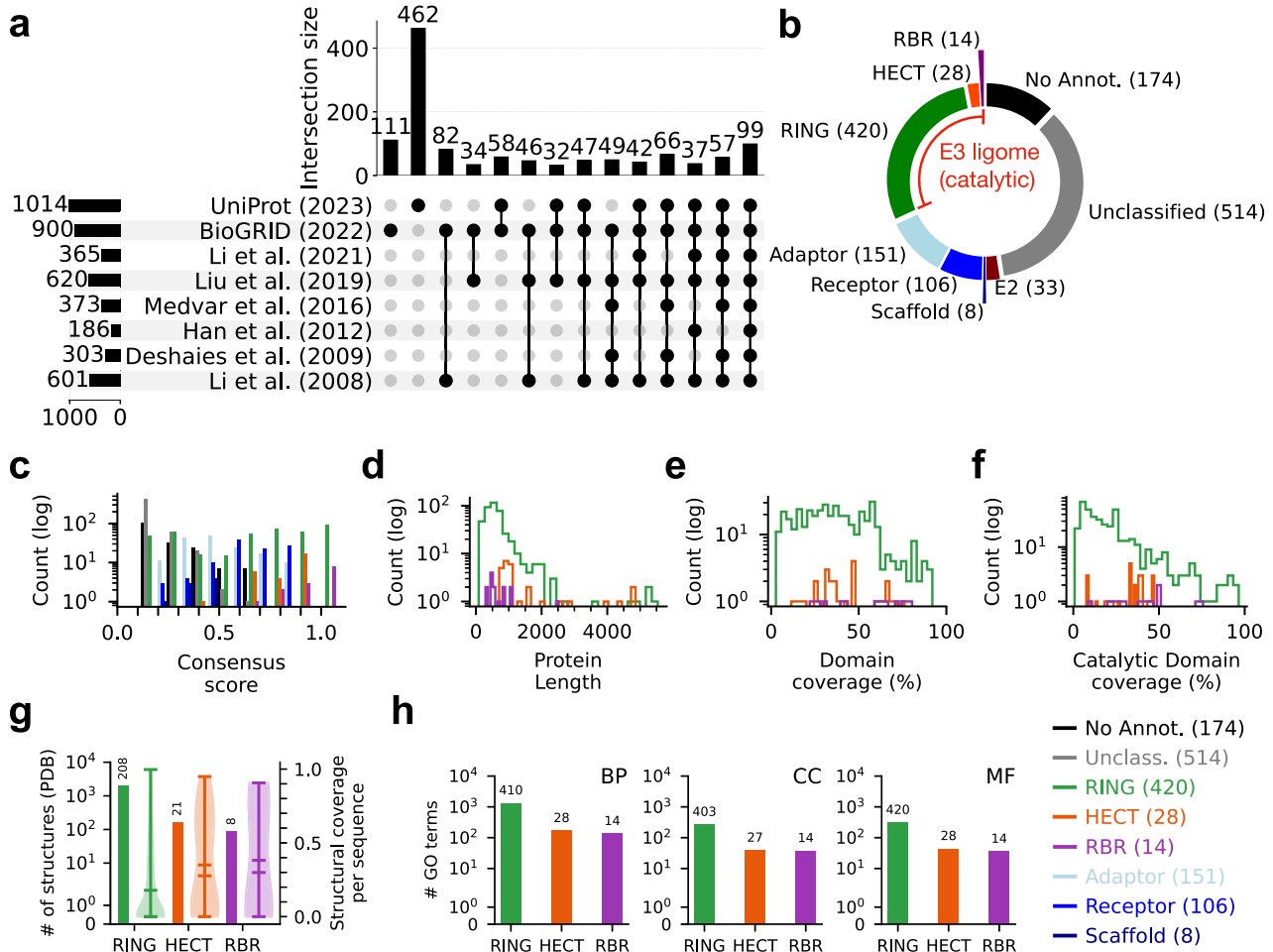

**Fig. 1 | Diversity of the human E3 ligome. a** A visualization showing the intersections of eight E3 ligases datasets ($A_1, \cdots, A_8$) obtained from existing literature and public repositories. The matrix layout for all intersections of individual datasets is sorted by size. Filled circles and their corresponding bars indicate sets that are part of the intersection and their sizes, respectively. Individual proteins ($X_i$) from the all eight datasets $\bigcup_{n=1}^{8} A_n = 1448$ annotated with one or more domains, $d_i$, belonging to a set of well-studied catalytic components of E3 enzymes ($C = \{d_c\}$) were compiled to form the high-confidence E3 ligome, $\{X_i \in \bigcup_{n=1}^{8} |\exists d_i \in C\}$. **b** Pie chart showing the extent of protein annotations and filtering to identify the catalytic and non-catalytic components of the human E3 ligome. **c** Distribution of consensus scores for all annotated protein classes reflects cross-dataset reproducibility on E3 ligase catalytic components. The distribution of (**d**) protein lengths and annotation coverage for (**e**) all domains and (**f**) catalytic domains highlights the heterogeneity of the E3 ligome. **g** Distribution of structural coverage of the E3 ligome at class-level. Barplots (left axis) display the number of available PDB structures for $n = 208$ RING, $n = 21$ HECT, and $n = 8$ RBR proteins. Violin plots (right axis, min, max, median, and mean values with mirrored density estimates on either side) represent distributions of fractional coverage for $n = 2001$ RING, $n = 168$ HECT, and $n = 90$ RBR structures. **h** The total number of unique GO terms associated with E3 classes indicates their functional vista under biological process *BP*, cellular component *CC*, and molecular function *MF* ontologies. *n*-values on bars indicate unique proteins with GO terms.

## Metric learning for classification of the human E3 ligome

To study the organization and relationships of proteins within the human E3 ligome, we attempted to classify these enzymes using multiple sequence alignment (MSA) followed by phylogenetic tree construction. However, we obtained a low-quality MSA with numerous gaps (Supplementary Fig. 2a), primarily due to (i) high sequence divergence, (ii) numerous proteins with uneven length distributions, (iii) inadequate alignment of conserved, catalytic domains, and (iv) an extensive repertoire of domain architectures (Supplementary Fig. 2b).

To capture the complex relationships within the human E3 ligome, we used a machine-learning approach to learn an emergent distance measure. Using a linear sum model, we combined multiple distance measures with optimal weights to reproduce class-level organization (partial ground truth) in hierarchical clustering (Fig. 2a). We first computed twelve pairwise distance matrices for all E3 ligase pairs ($d_{PQ}^i$ where $i = \{1, \cdots, 12\}$, for all P and Q $\in$ E3 ligome; $12 \times {}^{462}C_2$ distances) across distinct granular layers: primary sequence, domain architecture, 3D structure, function, subcellular localization and expressions (see "Methods"). These distances between ligase pairs are widely distributed and capture their relationships across distinct molecular- and systems-level hierarchies (Fig. 2b). Interestingly, most distance measurements showed low correlations (Fig. 2c), suggesting that they capture largely orthogonal information from the distinct granularity layers. Only the three domain architecture-based distances which quantify domain composition ($d_{PQ}^{Jac}$), domain order ($d_{PQ}^{GK\gamma}$), and domain duplication ($d_{PQ}^{Dup}$) are highly correlated (Pearson $r \geq 0.5$). Further, the 3D structure-based distance measure ($d_{PQ}^{Str}$) is also positively correlated with domain composition and duplication distances (Pearson $r \geq 0.5$).

Next, to learn an emergent distance measure, $D_{PQ}$, we combined four individual distances ($d_{PQ}^i$), representative of E3 sequence, domain composition, structural, and functional level organization, with their appropriate weights ($w_i \in \{0.05, \cdots, 0.95\}$ in 0.1 intervals).

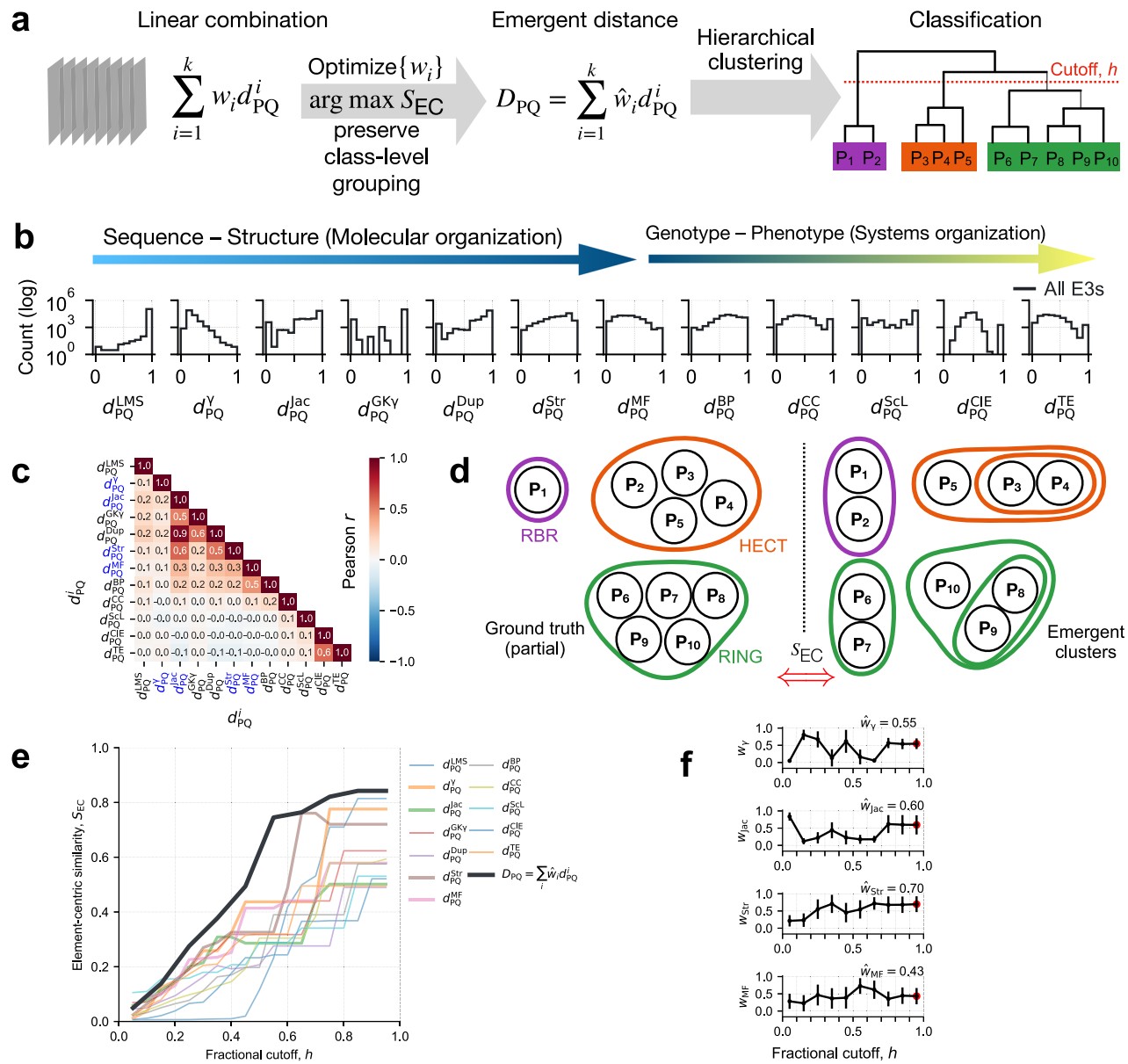

**Fig. 2 | Metric learning for E3 ligases. a** Schematic of the metric learning process. **b** Distribution of various pairwise distance measures spanning the molecular and systems level organization. **c** Pearson correlation of distance measures indicate orthogonality, mostly $r \in (-0.3, 0.3)$. Distances based on sequence alignment, domain composition, 3D structure (catalytic), and molecular function (marked in blue) are combined into an emergent distance ($D_{PQ}$) with appropriate weights. **d** By maximizing element-centric similarity, a measure of the overlap of emergent hierarchical clusters (right) with the ground truth (left) (**e**) evaluates individual metrics and their linear combinations. **f** Regression weights (mean± S.D.) corresponding to the four relevant distances as a function of fractional tree cutoff $h$. 100 clusters with largest $S_{EC}$ were sampled at each value of $h$ to estimate the mean and S.D.

These distances capture intrinsic molecular attributes and their relationships spanning the molecular scale. By uniformly sampling the weights, we constructed $10^5$ combination measures as a function of the hyperparameter (fractional tree cutoff, $h$, between 0.05 and 0.95). By simultaneously maximizing element-centric similarity[22] of the emergent hierarchical clusters resulting from combined measures, with partial ground truth (weakly-supervised scheme, Fig. 2d), we optimized an emergent distance measure ($D_{PQ}$) with appropriate weights ($\hat{w}_i$). We found that the linear combination of distances provided clusters with high element-centric similarity $S_{EC}$ compared to clusters obtained from individual distances (Fig. 2e, black curve vs. colored).

Normalized Mutual Information (NMI) and Fowlkes–Mallows Index (FMI) compare clustering assignments (various distance-

based vs. ground truth), but they are sensitive to cluster count (determined by tree cutoff, $h$; Supplementary Fig. 3a). Therefore, optimized weights $\hat{w}_i$ were obtained by averaging one hundred realizations of hierarchical clustering with maximum $S_{EC}$[22]. The weights corresponding to maximum $S_{EC}$ initially varied and then plateaued (at $h \geq 0.75$; Fig. 2f), resulting in the construction of an optimized emergent distance measure, $D_{PQ}$ (Eq. (1)). We found that the relative influence of 3D structure, domain composition, and sequence alignment ($\hat{w}_i \geq 0.5$) was more significant on the final learned metric and its ability to reproduce class labels accurately. Compared to the emergent distance measure, we found variable tree topologies with poor overlap and highly entangled trees for all four individual distances (Supplementary

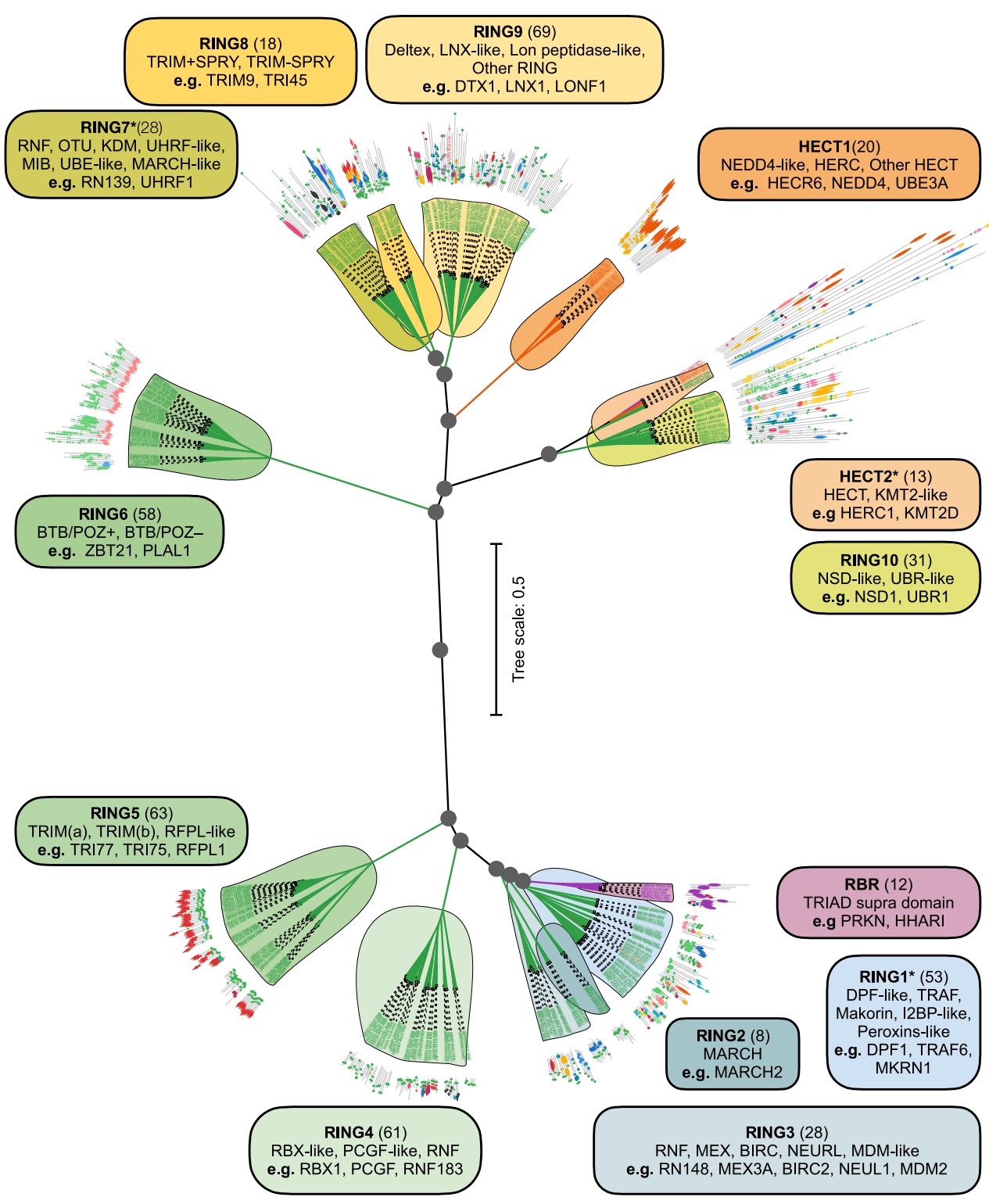

**Fig. 3 | Classification of the human E3 ligome.** Unrooted hierarchical tree computed using the optimized emergent distance metric $D_{PQ}$ (scaled branch lengths). The RBR (purple), HECT (orange), and RING classes (blue/green/yellow) are partitioned at $h = 0.25$ into 1, 2, and 10 families, respectively. Each cluster is defined by shared sequence, domain-architectural (mapped), structural, and functional elements. Boxes show family information, i.e., family name, size, and subfamilies, with representative examples. Grey-filled circles denote bifurcation nodes with ≥ 95% bootstrap support, and * denotes families with a few class-level outliers (3/13).

Figs. 3b–e).

$$D_{PQ} = 0.43 d_{PQ}^{MF} + 0.55 d_{PQ}^{\gamma} + 0.60 d_{PQ}^{Jac} + 0.70 d_{PQ}^{Str}. \quad (1)$$

## Organization of the human E3 ligome

Using the optimized emergent distance metric, $D_{PQ}$ (Eq. (1)), we constructed a scaled hierarchical tree classifying the human E3 ligome (Fig. 3 and Supplementary Fig. 4a). To assess the validity of nodes, branch stability, and the robustness of our classification, we resampled

the emergent distance matrix ($n$ = 500) and assigned bootstrap support at each branch point (Fig. 3, grey circles). The bootstrap support for all nodes beyond tree cutoff, $h > 0.15$, is 95–100%, indicating a stable branch pattern (Supplementary Fig. 4b) with a fixed tree topology. At $h \leq 0.15$, the bootstrap support for the nodes dropped drastically. This allowed us to use a tree cutoff threshold, $h = 0.25$, to parse the dendrogram and obtain robust and stable clusters with clear family and subfamily patterns while preserving RING-, HECT-, and RBR-class segregation.

We identified thirteen distinct clusters or E3 families ($h = 0.25$). At the class level, the E3 ligome is well segregated into ten RING families (Fig. 3, blue to green colors; clockwise arrangement from RING1 to RING10), two HECT (Fig. 3, top-branch; orange), and one RBR family (Fig. 3, bottom-branch; purple). Each E3 family is subdivided into one or more subfamilies (Fig. 3, boxes) with distinct patterns. Mapping domain architecture information onto the individual leaves aids recognition of well-preserved sequence and domain features, consistent with family and subfamily grouping, a pattern more evident in the unscaled circular dendrogram of the E3 ligome (Supplementary Fig. 4a). Further, few heterogeneous families are grouped more closely and emerge from single branches (bootstrap support ≈ 90−95%; Supplementary Fig. 4b) hinting at divergence of plausible superfamilies: (i) RBR and RING1–3 branch (small E3s), (ii) RING7−9 branch (medium E3s), and (iii) HECT2−RING10 branch (large E3s). This organization stems from the central node that bifurcates the E3 ligome into two groups characterized by average protein size (Fig. 3). The bottom branch displays six families with smaller E3s, while the top branch groups seven larger E3 families.

The E3 family organization reflects mechanistic differences (Supplementary Fig. 4c and Supplementary Table 3). The RING E3s mediate the direct transfer of Ub to the substrate, while the RBR and HECT E3s enable ubiquitin transfer via a two-step mechanism. The RBR-containing E3s form a homogeneous cluster, highlighting their conserved sequence and the TRIAD supra domain. Similarly, HECT-domain-containing E3s are organized into two clusters/families, HECT1 and HECT2. The HECT1 family is homogeneous and includes three subfamilies: NEDD4-like, HERC, and other HECT E3s. The HECT2 family contains a pure HECT E3 subfamily and an outlier subfamily containing large multi-domain RING-type E3s ( > 2000 residues) often with atypical mechanisms (e.g., MYCBP2, RNF213, see Supplementary Note 2). The most abundant RING-domain-containing E3s are organized into 10 families, each characterized by further grouping related proteins into distinct subfamilies with shared sequence elements, domain architectures, and structural features. For instance, the RING2 family comprises membrane-associated RING-CH-type domain (MARCH) E3 ligases (Fig. 3, bottom-right). This family includes all small MARCH E3 ligases characterized by their transmembrane domains and sequence lengths below 500 amino acids. TRIM E3 ligases are exclusively limited to two distinct families, RING5 and RING8, and feature the SPRY domain (Fig. 3, bottom-left). E3 ligases containing BTB/POZ and Zn-finger domain repeats are grouped into the RING6 family (Fig. 3, upper-left).

Although our emergent metric largely maximizes pure and homogeneous clusters (e.g., RBR, RING2, RING5, RING6, RING8, and HECT1), heterogeneity often arises at the subfamily level, resulting in sub-groupings of E3s with varied and unique domain architectures. Isolated proteins (singletons) in the RING1, RING7, RING8, and RING9 families form distinct subfamily groupings, complicating pattern detection. Only RING1, RING7, and HECT2 families display occasional class-level outliers (Supplementary Table 3). Supplementary Notes 3 to 15 describe each family structure in detail with information on sub-family branching, characteristic features, and distinct patterns, along with outliers providing a nuanced description (Supplementary Figs. 5–17 and Supplementary Notes 3–15).

## Functional segregation of the human E3 ligome

To evaluate the human E3 ligome, we conducted a CRISPR-Cas9 dropout screen of UPS genes, using cellular fitness as the main phenotype. We identified 53 catalytic and 32 non-catalytic E3 components to be essential for cell fitness (FDR ≤ 0.05 and $|\log_2(FC)| \geq 1.0$; Fig. 4a, b). Notably, these essential E3s were predominantly enriched in RING1/4/7, and RING9 families, suggesting critical biological roles (Fig. 4a). Several E2 enzymes and adaptors were also essential, reinforcing the importance of the ubiquitin conjugation and multi-subunit E3s (Fig. 4b). Overall, our CRISPR screen replicates correlated well with DepMap data (Pearson $r \geq 0.5$; Supplementary Fig. 18a). GO analysis of 53 essential E3s showed significant enrichment for nuclear components and DNA damage, replication, and repair processes (Fig. 4c), indicating their roles in genome integrity and nuclear regulation fundamental to cell survival. These findings point to essential E3 components crucial for cell viability.

To understand the functional diversity of the human E3 ligome, we filtered high-confidence GO terms and mapped them onto our classification, enabling us to draw functional clusters and visualize their networks across all three ontologies. This allowed recognition of generic and family-specific functions (enriched, $-\log_{10}(p) \geq 2$). At the BP level, as expected, the network analysis revealed prominent core functional sub-clusters associated with all terms containing "ubiquitination (Ub)" (Fig. 4e, Top). These BPs are shared across all families, indicating their generality. Additional core clusters relate to innate immunity, host-driven viral restriction, NF-κB regulation, and IL-17 signaling (Fig. 4d). Further, cooperative diverse non-degradative functions such as regulation of gene expression, protein stability, cell growth, and ERAD pathway are enriched within the E3 ligome (see Supplementary Note 16).

Examining family-specific GO enrichment uncovered functional specialization supported by experimental evidence (Fig. 4e, colored triangles; Supplementary Table 4). For instance, the RBR family members, RNF14, RNF144A, and PRKN, demonstrated specificity for K6-linked-Ub (Fig. 4f, left). K6-linked chains flag stalled RNA-protein cross-linked complexes (RNF14), DNA-sensing adaptor STING for activation of interferon signaling (RNF144), and damaged mitochondria for clearance (PRKN)[23–25]. Similarly, TRIM E3s (RING5) were significantly enriched in antiviral innate immune response (Fig. 4f, right). They regulate pattern recognition receptor activity in cells, such as RIG-1 and MDA5-mediated responses[26,27].

To test family-specific functions, we performed whole-cell proteomics in response to cellular perturbations (EBSS and CPT treatment) and monitored the differential expression of E3 ligases. The cellular responses to EBSS and CPT treatments are broad and multifaceted, impinging on a wide range of BP functional clusters (Fig. 4e, blue and orange triangles). 18/34 implicated E3s have direct evidence linking them to starvation response pathways (Supplementary Table 4). Notably, MGRN9 and BRCA1 were also essential in our CRISPR screens (Fig. 4g, left). Our analysis revealed a differential expression ($p$ value ≤ 0.05) of the E3 ligases TRIM27/32, and UBR1 in addition to key autophagy regulators SQSTM1, CALCOCO2, GABARAPL2, and MAP1LC3B2, highlighting a coordinated modulation of the autophagic machinery during starvation (Fig. 4g, right colored vs. blue). In contrast, the observed up-regulation of EIF4EBP1, a translational repressor regulated by mTOR signaling[28], and GDF15, a stress-responsive cytokine involved in metabolic adaptation[29], indicates activation of complementary stress response pathways that may support cellular survival. TRIM27/32 regulate autophagy initiation and selective degradation pathways by ubiquitinating essential autophagy proteins such as ULK1 and p62, thereby promoting autophagosome formation and cargo recognition[30–32].

Similarly, experimental evidence links 29/87 implicated E3s directly to DNA damage response (DDR) (Supplementary Table 4), of which 12 E3s turned out to be essential (e.g., TRAF3/7, MDM2; Fig. 4h,

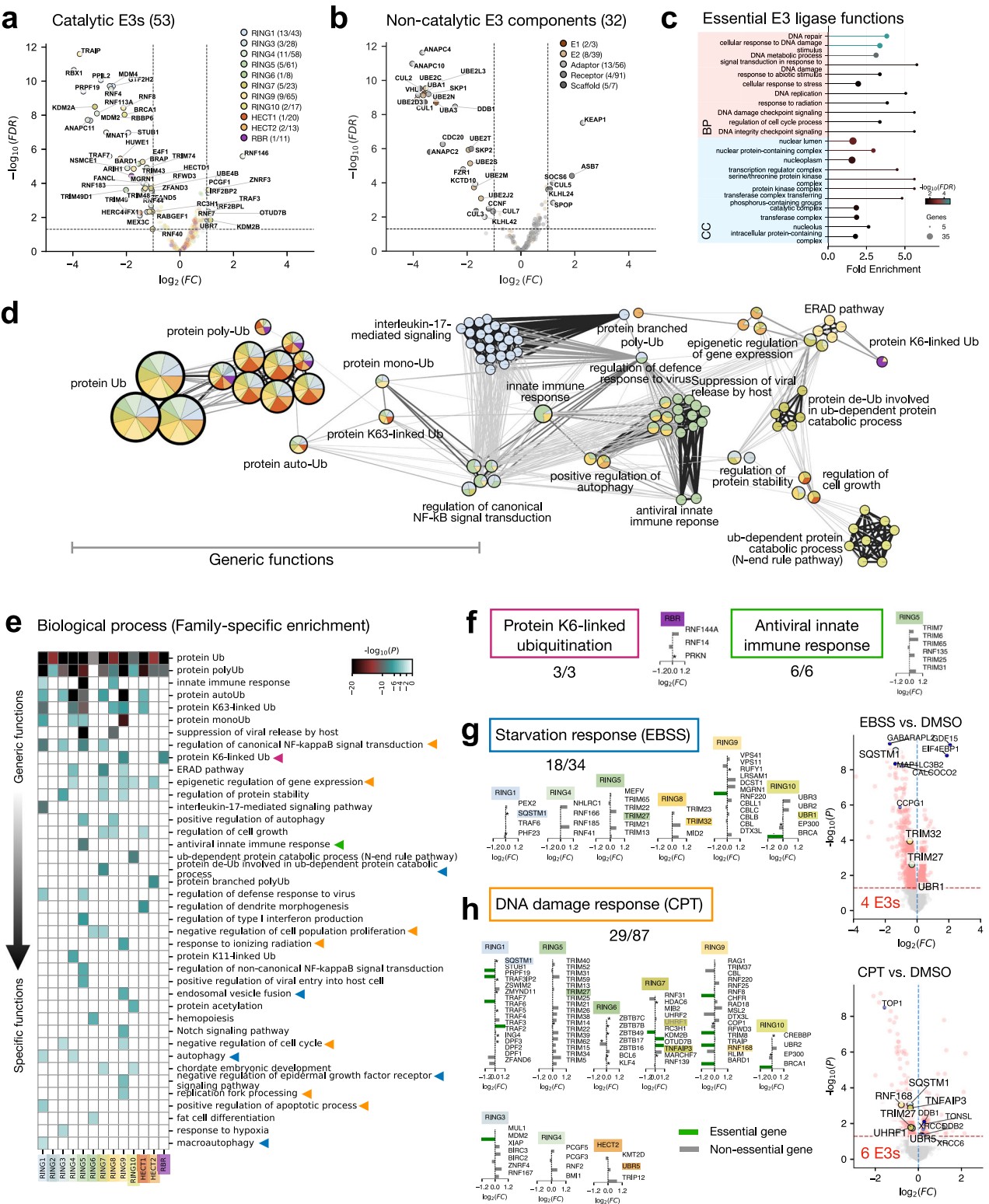

left). We found that the E3 ligases TRIM27, UHRF1, TNFAIP3, and RNF168 were significantly down-regulated in response to CPT, while UBR5 was up-regulated in addition to the control proteins (e.g., TOP1, XRCC6, DDB1; Fig. 4h, right, colored vs. blue). TOP1, targeted by CPT, forms covalent DNA-protein cross-links that cause replication-associated double-strand breaks and trigger DDR[33]. In response, ubiquitin signaling, via RNF8/168-mediated K63-linked histone-Ub, recruits DDR mediators like 53BP1, BRCA1, and RAD18[34,35]. UBR5 limits Ub-signaling by degrading RNF168, ensuring DNA repair fidelity.

Further, UHRF1 promotes DDR by driving chromatin remodeling and BRCA1 recruitment[36]. Additionally, UHRF1, RNF168, and RNF8 function in DNA replication[37]. Together, these mechanisms underscore Ub-signaling to maintain cellular homeostasis and genome integrity.

At the MF level, all E3s have "ubiquitin-protein ligase" activity (Generic; Supplementary Fig. 18b). More than 20 MFs could be attributed to family-specific domain architectures (Supplementary Fig. 18d). The Zn-finger domains are also common to transcription factors. For e.g., they equip E3s for p53 binding (RING3), histone-Ub

**Fig. 4 | Functional segregation of the E3 ligome.** Volcano plots of Gene essentiality analysis derived from CRISPR screens for (**a**) catalytic and (**b**) non-catalytic components of the E3 ligome. **c** GO enrichment analysis for essential catalytic E3s. **d** The functional landscape of the E3 ligome (biological processes) is captured by the network of GO annotation clusters. Individual nodes representing GO clusters (20 labeled) are drawn as pie charts (size proportional to # of E3s; colored by family enrichment) connected by distinct edges (κ-similarity ≥ 0.3). **e** The heatmap displays all functional clusters corresponding to family-specific enrichment of E3s (*p* value estimated using hypergeometric test (two-sided), discrete color scale for *p* value ≤0.01; white otherwise). Colored triangles show examples of family specific enrichment for (**f**) K6-linked ubiquitination (purple) and antiviral innate immune response (green), (**g**) starvation response under 6h EBSS treatment (blue), and (**h**) DNA damage response under 4h 100 nM CPT treatment (orange). For panels **f**–**h** gene essentiality data $\log_2(FC)$ or DepMap Gene Effect scores (*) are plotted for individual E3s. The ratio denotes the fraction of E3s with experimental evidence (PMIDs) for GO functions directly. **g, h** panels also show volcano plots of proteomic analysis, highlighting significantly up-regulated and down-regulated proteins (red scatter; adjusted *p* values were obtained using Benjamini-Hochberg method in two-sided moderated t-tests) with overlapping E3s (colored) and control proteins (blue filled circles).

(RING9), and unmethylated CpG binding (RING7). Other non-catalytic E3 domains mediate PPI interactions with actin, SH3-domains, Kinases, and proteases. Distinct subcellular localization of E3s exerts spatial control of protein-Ub (Supplementary Fig. 18c). Most E3 ligases are cytosolic, which form an essential part of the "ubiquitin ligase complexes" (CC: Generic). We find enriched E3 families for distinct CCs: CD40 receptor, GID, and nBAF complexes (RING1); early endosomes and lytic vacuoles (RING2). In total, we identified 17 unique cellular components with distinct E3-specific enrichment patterns (Supplementary Figs. 18e, 19a).

GO annotations from author/curator statements and electronic methods—despite varying confidence levels—provide a rich, abundant dataset of key testable hypotheses on E3 systems (Supplementary Figs. 18f, 19a, and Supplementary Note 17). Publication counts for each protein-specific annotation highlight knowledge gaps and pinpoint well-studied and underexplored E3 systems (Supplementary Fig. 19b).

## Interaction landscape of the human E3 ligome

E3 ligases can operate as standalone or multi-subunit enzymes. Complex E3s consist of scaffolds, adaptors, and substrate receptors that determine specificity, stability, and regulation[21]. For instance, RBX1 binds scaffolds (CUL1–CUL5) and anchors the E2 enzyme to form the catalytic core for Ub transfer (Fig. 5a). Its interactions with various cullins, adaptors, and receptors enable ~250 CRL configurations, providing modular regulation and substrate specificity. By contrast, standalone E3 ligases, like MDM2, c-CBL, PARKIN, or SMURF1/2, either have specialized domains or undergo specific PTMs that recognize substrates and facilitate E2 binding and ubiquitin transfer. For example, HECTD3 operates via a two-step ubiquitin transfer mechanism (Fig. 5b). However, substrate binding occurs through specific motifs within the non-HECT regions that presumably recognize particular degrons (sequence motifs, distinct PTMs, or unique structural elements).

Previous annotations[38,39] report 6 complex, 329 standalone, and several unclassified E3s. We extended this annotation by curating non-catalytic components and cataloging their direct interactions with E3s (Fig. 5c). Multi-subunit complex structures are only resolved for four E3s (RBX1/2, ARI1, and APC11). While partial complexes are resolved for 12 E3s (e.g., APC11, ARI1/2), we found several binary direct interactions between E3s and non-catalytic subunits, re-annotating 75 E3s operating as complexes (Fig. 5d, black), leaving 277 standalone[38] and 110 unclassified E3s (Fig. 5d, red). RING8 family displayed many complex E3s (50%), followed by RING1 (26%), while RING2 and HECT2 families displayed entirely standalone E3s (Fig. 5e, Supplementary Table 5). Consistent with our findings, we observe that MARCH-type E3s (RING2) operate in the membrane environment primarily as standalone enzymes. HECT2 proteins (e.g., HECD3) possess multiple domains for adaptor, receptor, and scaffolding, explaining their standalone function.

Next, we assembled the E3–substrate interaction (ESI) network by integrating data from known ESIs (*n* = 2012), direct PPIs (*n* = 5844), indirect PPIs (*n* = 6530), and predicted ESIs (*n* = 64802; Fig. 5f, see Methods). Integrating these data and verifying their ubiquitination status resulted in excluding false positives (E3-associated) and

improving the annotation of likely substrates (Supplementary Figs. 20a–c). This enabled mapping ≈ 62% (*n* = 7691 substrates) of the ubiquitinated human proteome (Fig. 5g).

Analysis of the E3–substrate network revealed distinct specificity patterns. Using well-known ESIs, we observed that most E3 ligases have a single substrate ($\sim 10^2$), fewer target multiple ($\sim 10^1$), and only a handful possess extensive substrate sets (Supplementary Fig. 20d). Most substrates are targeted by E3s belonging to two or more families (*n* = 5520 Promiscuous substrates, *n* = 72749 interactions; Fig. 5h, bottom; Supplementary Table 6). We also identified substrates that are potentially ubiquitinated by two or more E3s belonging to the same E3 family (*n* = 804 Family-specific substrates, *n* = 3292 interactions; Fig. 5h, middle) and substrates uniquely targeted by specific E3 ligases (*n* = 1367 E3-specific substrates, *n* = 1367 interactions; Fig. 5h, top).

For instance, the E3 ligase SMUF1 specifically targets TBX6 for degradation during cell differentiation[40]. Similarly, MARCH5 specifically targets FIS1 for ubiquitination (Fig. 5i) to regulate mitochondrial fission[41]. Both NEDD4 and ITCH belong to the HECT family and ubiquitinate MART1 to exert complementary functions for the sorting and degradation[42], and PACS2 is ubiquitinated by BIRC2 and BIRC3, members of the RING3 family (Fig. 5i), conferring TRAIL resistance to hepatobiliary cancer cell lines[43]. CDN1A, a central regulator of cell cycle and DNA damage response, is ubiquitously targeted by diverse E3 ligases, linking multiple signaling pathways to replication checkpoints (Fig. 5i). Overall, integrating the ESI network obtained here with ubiquitination and degron mapping efforts provides a powerful framework to uncover additional E3-substrate relationships, validate candidate ligases, and reveal functional redundancies or specificity (Supplementary Note 17).

## Druggability map of the human E3 ligome

To learn likely avenues of proximity-based therapeutics and leverage the relationships within the human E3 ligome, we mapped existing E3 handles derived from known Proteolysis Targeting Chimeras (PRO-TACs) and E3 binders to individual E3s and their families (Supplementary Fig. 21a, Supplementary Table 7). Only 16 proteins (9 catalytic E3s and 7 adaptors) are directly targeted by existing E3 handles (Fig. 6a, top). A large fraction of the designed E3 handles are specific to adaptor proteins (e.g., VHL, CRBN), and only a very select few directly target the catalytic E3s (e.g., XIAP, MDM2/4/7, BIRC2/3/7; Supplementary Fig. 21b). Nearest neighbor analysis using our E3 ligome identified five closely related proteins (BIRC8, RN166/181/141, and UBR2; Fig. 6a, top; grey boxes). Given their high structural similarity (often paralogs), the same E3 handles could be repurposed to target them. Mapping small-molecule E3 binders gave us a potential set of compounds targeting 25 additional E3s and 15 non-catalytic components, thus identifying unexplored targets and avenues for lead development for the rational design of E3 handles (Fig. 6a, bottom; red labeled).

Next, we mapped the chemical landscape of E3 handles and E3 binders using the Uniform Manifold Approximation and Projection (UMAP) embedding of high-dimensional 2048-bit Morgan fingerprints. By clustering them, we visualized their molecular similarities. Comparison with other dimensionality reduction methods, such as PCA and

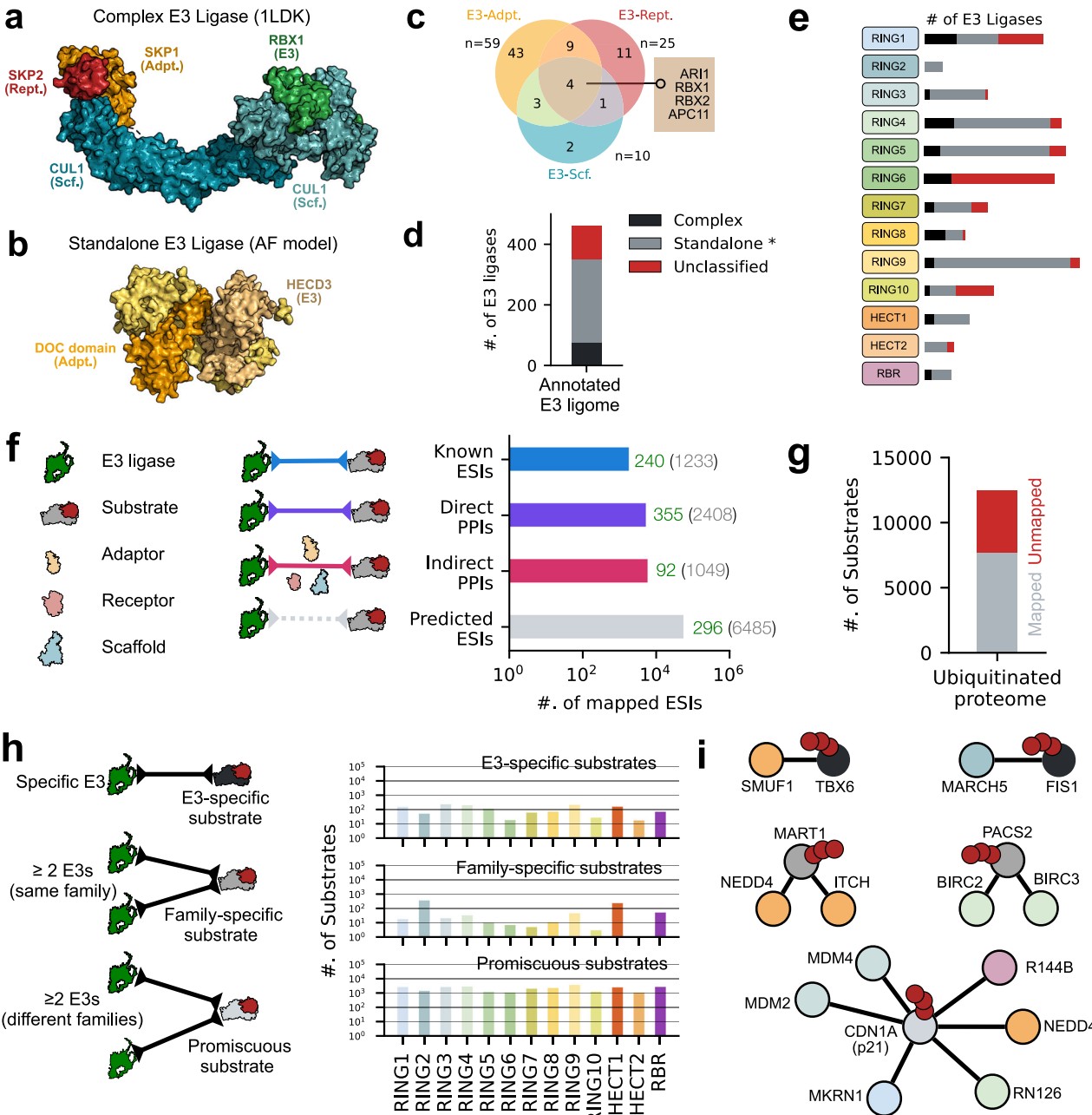

**Fig. 5 | Protein−protein interactions of the E3 ligome.** Representative examples of E3 ligases functioning as a (**a**) multi-subunit protein complex (CRL) or (**b**) a standalone enzyme (HECD3). **c** Venn diagram of pairwise interactions of adaptors, receptors, and scaffold proteins with E3s. **d** Annotation of 462 E3 ligases into complex, standalone, or unclassified modes of action. **e** Family-wise mapping of data from (**d**). **f** Pairwise E3−substrate interactions for all E3s obtained by integrating data from known ESIs, mapped transient direct and indirect PPIs, and predicted ESIs. **g** Mapping of the ubiquitinated proteome with E3s ($\approx 62\%$, $n = 12464$). **h** Schematic showing substrate categorization into E3-specific, family-specific, and promiscuous classes (left) and their relative distributions mapped onto E3 families (right). **i** Representative examples for the three types of ESI networks.

t-SNE, did not provide optimal clusters (Supplementary Figs. 21c−d). We detected 20 chemically distinct clusters within the UMAP space (Fig. 6b, indexed, distinct colors) by fast search and identification of density peaks[44]. 12 compound clusters contain only E3 binders, while the remaining 8 also contain E3 handles targeting specific proteins (Supplementary Fig. 22). The top 6 clusters occupy a large central sub-space, indicating cluster heterogeneity (relatively large chemical space). These compound clusters predominantly target proteins from RING3 and RING7 and adaptors (9123 molecules, 67%). E.g., the cluster #2 contains 1222 ligands forming 5 dense sub-clusters representing distinct chemo-types that selectively target MDM2/4, IRAK4, XIAP,

HDAC6, and BIRC2, respectively (Fig. 6c). Further, the compounds from clusters #1−6 target a large number of proteins and display less specificity (Fig. 6d; $LP_{ij} \leq 3$, low-binding likelihood). The clusters #9−20 are relatively more homogeneous, occupy small peripheral regions of the UMAP-space, and display high selectivity (1−3 targets/cluster with $LP_{ij} \geq 3$, cyan squares).

3D structures of compound−E3 interactions are limited. Our mapping showed that only 231/14615 pair-wise interactions are resolved in the PDB (Fig. 6e, covering 212/13620 compounds). These structures mainly feature well-studied E3 handles and binders, high-lighting clear examples for drug repurposing. In cluster #13, the

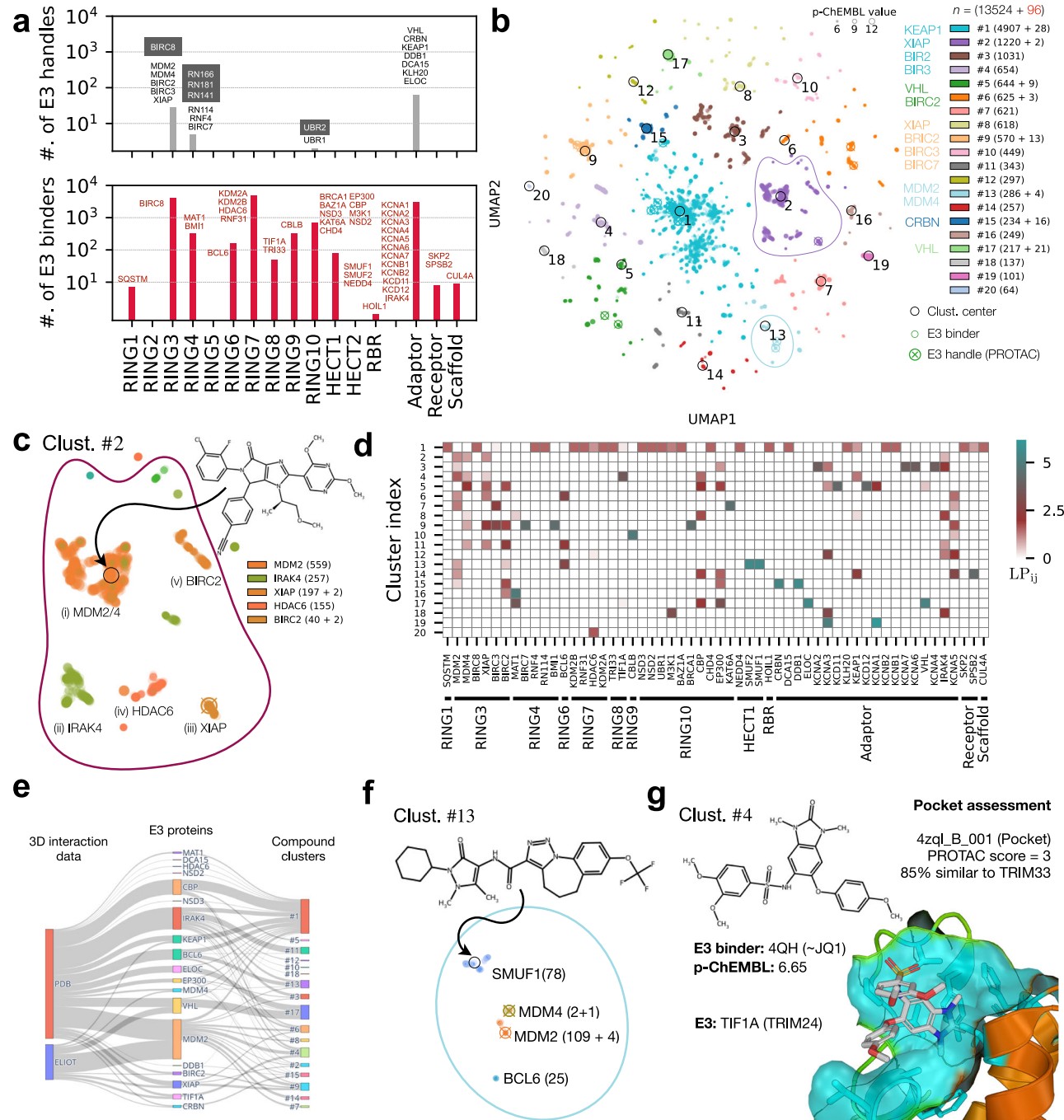

**Fig. 6 | Druggability map of the E3 ligome. a** Distribution of known E3 handles (extracted from PROTACs, top) and expanded set of E3 binders (potential lead compounds, bottom) targeting E3 families. Individual proteins uniquely targeted by E3 handles (n = 16, black) and E3 binders (n = 40, red) are displayed for each family. Grey-filled boxes (top) show closely related protein targets for E3 handle/ PROTAC repurposing. **b** Reduced 2D UMAP chemical space of E3 handles (n = 96) and E3 binders (n = 13524); size proportional to p-ChEMBL value. Compound clusters (colored) within UMAP space represent distinct chemical structures (Cluster centers indexed #1–20) are identified by local density peaks (see Supplementary Figs. 21). **c** Magnified view of cluster #2 showing dense sub-clusters of compounds targeting multiple proteins. **d** Log-transformed propensities, $LP_{ij}$ of

individual compound clusters capture binding likelihood. **e** Sankey plot showing the map between PDB (3D interaction), ELIOT (pocketome), individual E3 proteins (19/56), and their compound clusters (covering 212/13620) from our small-molecule interaction analysis. **f** Magnified view of cluster #13 showing the proximity of E3 handles targeting MDM2/4 to SMUF1/BCL6 binders, inferred from compound similarity and clustering in the reduced UMAP representation. **g** Example of a potential lead compound identified from cluster #4. Ligand 4QH (similar to JQ1) binds to TIF1A bromodomain (PDB code: 4ZQL) and can be developed into a specific E3 handle. Binding site analysis (from ELIOT) indicates a favorable PROTAC score and high similarity to the TRI33 bromodomain pocket (RING8 member).

MDM2/4-specific E3 handles could be repurposed to target SMUF1 (HECT) or BCL6 (RING6). The presence of SMUF1 binders or BCL6 binders, forming dense sub-clusters nearby, reflects their chemical similarity (Fig. 6f). Further, analysis of surface pockets of E3s could

provide structural context and reveal similarity patterns for the development of lead compounds into E3 handles. To this end, we integrated the ELIOT data[45], which includes detailed pocket analysis for 73 compounds bound to 8 E3 ligases from our small-molecule

interaction dataset ($n$ = 74, Fig. 6e). Interestingly, in cluster #4, we found a compound, 4HQ, bound to the bromodomain of TIF1A (Fig. 6g). Given its moderate affinity and favorable ELIOT pocket assessment, it could be developed into a specific E3 handle for either TIF1A or homologous TRI33 (85% pocket similarity). Moreover, 4HQ is similar to JQ1, a well-known ligand targeting the bromodomain of human BRD4 (PDB code: 3mxf). Further, integration and comparison of sub-cellular location, cell- and tissue-specific expression patterns of E3s will be essential for effective drugging of the human E3 ligome (Supplementary Fig. 23). In summary, this approach enables rational and family-informed PROTAC design by linking chemical tractability with context-aware E3-substrate relationships (Supplementary Note 17).

## Discussion

Navigating the vast and complex landscape of E3 ligase biology requires a comprehensive approach. Despite decades of dedicated investigation, the intricate diversity and functional complexity of E3 ubiquitin ligases continue to pose a significant challenge. In decoding this complexity, we first curated and filtered E3 ligases, ensuring data accuracy, consistency, and relevance for all downstream analyses. By assigning confidence scores to each ligase and employing stringent inclusion criteria, we remove false positives and improve annotation, providing a high-quality and comprehensive human E3 ligome. Ultimately, this simplification facilitated the identification of key catalytic components and paved the way for applying machine learning and algorithmic approaches to E3 systems.

The human E3 ligome exhibits remarkable heterogeneity, evident in its diverse sequence, domain architectures, structures, and functions. This diversity is shaped by not only the evolutionary forces influencing domain shuffling and genetic rearrangements but also biophysical forces influencing molecular recognition and spatio-temporal regulation of enzymatic reactions, leading to specialization and adaptation[46]. To effectively categorize E3 ligases, we require overarching organizational principles delineating broad evolutionary clans and functionally distinct subgroups within the E3 ligome. Hierarchical classification captures organizational principles, achieves higher prediction accuracy, and can handle previously uncharacterized data and class imbalances more effectively[47]. These methods enable a more precise and context-aware organization of proteins, facilitating the recognition of salient and unique features[48]. However, its performance heavily depends on choosing an appropriate metric reflecting authentic relationships.

Assessments of similarity and distance are critical components of human cognitive function and constitute a foundational element in developing machine learning and applying data mining techniques[49]. Using a weakly supervised learning paradigm, we optimized a linear metric that is simple, scalable, and straightforward to interpret with broad applicability. We bridged the molecular scale from protein sequence, domain architecture, 3D structure, and molecular function, resulting in a unique measure capable of detecting subtle shifts, reproducing class-level grouping of E3s, and improving family and subfamily definitions.

We present a multi-scale classification model enabling a comprehensive analysis of the E3 systems. We classify the E3 ligome into 13 distinct E3 families characterized by shared domains, comparable architectures, and similar 3D structures. By moving beyond traditional taxonomic methods and subjective, ad hoc classifications, our multi-scale model provides a systematic and objective framework. Although not explicitly dependent on any individual distance measure, it is strongly associated with shared structural similarities and domain architectures, providing exceptional resolution into functional specialization and mechanistic action of E3s. The largest RING class, divided into 10 families, shows remarkable diversity. For instance, MARCH, TRIM, and BTB groupings reflect domain architecture. The RBR class exhibits striking homogeneity, suggesting strong evolutionary conservation[50]. The HECT class is split into two individual families (HECT1 and HECT2), consistent with the previous classification[51]. These organizational insights lead to interesting additional hypotheses, revealing unexplored roles for existing E3s.

The functional diversity of the E3 ligome serves as a central hub for cellular homeostasis, extending far beyond the classical proteasomal degradation. CRISPR-Cas9 dropout screens identified a substantial number of catalytic E3 ligases and non-catalytic components as essential for cellular viability. They predominantly contribute to critical cellular processes, including DNA replication, repair, and maintenance of genome stability. Given the limitations of experimental data on E3 functions and the inherent challenges in quantifying them, GO terms serve as functional proxies. We find that the generic functions of E3 ligases are conserved across all families, encompassing protein ubiquitination, modification, and degradation (BPs); localization to E3 complexes or the cytosol (CC); and catalytic activity driving ubiquitin transfer (MF). More importantly, our classification captures functional segregation of E3 systems, providing significant insights into the distinct biochemical and functional mechanisms regulated by individual families. In test cases, we highlight how the RBR family promotes K6-ubiquitination[23], and RING5 E3s are involved in innate immune responses[26]. By integrating known experimental evidence, gene essentiality data, and proteomic profiling of HeLa cells, we mapped E3 systems involved in global starvation[30] and DNA damage response[52,53]. Furthermore, our quality analysis identifies additional testable hypotheses (from automated GO terms) and catalogs undercharacterized ligases.

Mapping the protein interaction landscape of the whole E3 ligome is challenging. We integrate disparate datasets to build enzyme–substrate network maps for each ligase family. We found that RING1/3/8 and RBR members contain more multi-subunit E3 complexes, while RING2 and HECT2 are likely to operate in a standalone manner, directly recruiting substrates. Further, we could classify substrate molecules into E3-specific, family-specific, and promiscuous substrates, providing foundational data for understanding the molecular principles of substrate recognition. Recognition of shared patterns in substrates can point to a better understanding of individual and group-specificity. Further, orthogonal data on subcellular localization of E3s and substrates, their tissue-specific expression patterns, explain the context-dependent ESIs and the prevalence of promiscuous substrates.

Targeted protein degradation via PROTACs is a promising therapeutic strategy to target previously undruggable proteome[54]. Despite its potential, progress in targeting unexplored E3s and the rational design of specific E3 handles has been gradual. Most often, PROTACs and glue-like compounds exploit ligands against well-known adaptor proteins like CRBN- and VHL-dependent modalities to target CRLs for the specific degradation of substrates. Only a few E3s have been directly targeted using PROTACs[55,56]. By leveraging the E3 ligome structure, we extend the map of E3 handles, increasing the likelihood of repurposing existing PROTACs to target closely related E3s in a family-specific manner. Further, by mapping an expanded set of E3 binders and associating them with unexplored E3s, we build a curated set of lead compounds with unique chemical signatures for further rational design of specific E3 handles. Although 2D-based UMAP visualizations offer a computationally efficient overview of chemical diversity and aid in hypothesis generation, they may fail to capture activity cliffs or bio-isosteric relationships. Complementing the druggability map generated here through physics-based approaches, including 3D conformational sampling and HTP docking screens of ligands against the druggable E3 pocketome[45] will propel the rational design of specific E3 handles. Furthermore, exploiting the emergent relationships offered by the E3 ligome, in combination with other metrics capturing their

localization, cell-, tissue-expression, enriched ESI networks, and a list of already targeted and unexplored E3 binders, allows an efficient drugging strategy for unexplored targets in health and disease.

In conclusion, the multi-scale classification framework developed here provides a comprehensive global view of the human E3 ligome. Mapping disparate multi-modal and multi-resolution data onto the ligome structure, such as functions, interactions, and druggability, provides a systems-level understanding, enabling high-throughput screening and profiling. The metric learning paradigm developed here is simple and transferable to other areas of data-driven biology. More specifically, the human E3 ligome offers clear actionable guidelines and a unified framework for systematic annotation and interpretation of ubiquitination data, enabling integration with high-throughput proteomics and genetic screens, including spatial proteomics to identify functionally relevant E3s, uncover pathway-specific patterns, and guide rational design of targeted degradation strategies. We anticipate that the data and insights presented here will stimulate further research into E3 systems and drive the development of innovative therapeutics.

## Methods

### Assembly of the human E3 ligome

We collected eight individual human E3 ligase datasets ($A_1, \cdots, A_8$) including previously published reports[17–19] and public repositories: E3Net[39], UbiHub[38], UbiNet 2.0[57], UniProt (retrieved on 2023-02-13 with keyword "e3 ubiquitin-protein ligase")[58], and BioGRID (retrieved on 2022-01-26)[59] compiled using multiple distinct criteria. We merged all of them to form an initial dataset ($|\bigcup_{n=1}^{8} A_n| = 1448$), visualized the overlap of individual resources using UpSet plot[60], and assigned a consensus score to each entry based on its presence/absence among the source datasets. We then compiled a list of distinct, well-studied E3 catalytic domains from InterPro[61] corresponding to RING, HECT, and RBR classes from all published sources ($C = \{d_C\}$). Using the presence of characteristic catalytic domain(s) $d_i$ within each polypeptide, we identified and filtered 1448 proteins corresponding to all catalytic subunits of E3 ligases, $\{X_i \in \bigcup_{n=1}^{8} | \exists d_i \in C\}$. This was followed by manual curation based on InterPro domain descriptions of possible catalytic activity (E2-binding and Ub transfer) to obtain the final refined set of 462 E3 ligases (E3 ligome).

In multisubunit E3 ubiquitin-ligases–most prominently the Cullin-RING ligases (CRLs)–three functionally distinct classes of subunits cooperate to bring an E2-Ub in proximity to a specific substrate. Large, rigid, central scaffold proteins (e.g., the Cullin family, Cul1–Cul5) organize the ligase complex by simultaneously binding the catalytic RING-finger subunit and the docking sites for adaptors or receptors. Adaptor proteins bridge modules that link a scaffold's N-terminal docking surface to a separate substrate-receptor. They often recognize a conserved motif on the receptor subunit (e.g., F-box, SOCS-box, WD40 proteins). The receptor proteins are specificity determinants that directly recognize and bind degron motifs on substrates. They define which substrates get ubiquitinated (e.g., Skp2, Keap1, VHL). We independently annotate and classify each of the three classes: 151 adaptors, 106 receptors, and 8 scaffold proteins, and use their PPIs to map distinct substrates for multi-subunit E3s.

### Multi-scale distance measures

We encoded the pair-wise relationship of E3 ligases by computing twelve distinct distances ($d_{PQ}$) spanning several granularity levels: primary sequence, domain architecture, tertiary structure, function, subcellular location, and cell line/tissue expression. All the distance measures were scaled between [0, 1] for comparison and even combination.

At the sequence level, we used an alignment-free local matching score-based (LMS) distance and an alignment-based $\gamma$ distance between protein pairs using the canonical isoform sequences. The LMS distance $d_{PQ}^{LMS}$ between two proteins P and Q is given by

$$d_{PQ}^{LMS} = 1 - \frac{2\,\mathrm{LMS}(P, Q)}{\mathrm{LMS}(P, P) + \mathrm{LMS}(Q, Q)}, \tag{2}$$

where $\mathrm{LMS}(P, Q) = \sum_{i \in \{P', Q'\}} M[i, i]$ captures the extent of local similarity by summing BLOSSUM62 substitution scores for overlapping 5-residue fragment pairs $\{P', Q'\}$ from proteins P and Q[62,63]. The pairwise $\gamma$ distance measures the evolutionary distance between the globally aligned sequences of two proteins, P and Q, where $p_{PQ}$ is the fraction of alignment positions with residue substitutions and indels, and $a = 2$[64].

$$d_{PQ}^{\gamma} = a[(1 - p_{PQ})^{-1/a} - 1], \tag{3}$$

To quantify the preservation of domain architectures among all protein pairs, we computed three distances: Jaccard, Goodman–Kruskal $\gamma$, and domain duplication distances, using domain annotations obtained from the InterPro database[61] (Nov 2022). The Jaccard distance[65,66] represents the compositional similarity of protein domains. It is the ratio of the number of shared ($N'_{PQ}$) and unique domains ($N'_P, N'_Q$) between proteins P and Q,

$$d_{PQ}^{Jac} = 1 - \frac{N'_{PQ}}{N'_P + N'_Q - N'_{PQ}}. \tag{4}$$

The Goodman-Kruskal $\gamma$ distance compares the order of domain arrangements between two proteins, P and Q, and is computed as

$$d_{PQ}^{GK\gamma} = 1 - \frac{1 + \gamma_{PQ}}{2}, \tag{5}$$

where $\gamma_{PQ} = (N_{PQ}^S - N_{PQ}^R)/(N_{PQ}^S + N_{PQ}^R)$ with $N_{PQ}^S$ and $N_{PQ}^R$ denoting the same- and reversed-ordered pairs of proteins P and Q, respectively[66,67]. Finally, the domain duplication distance[66] compares the overlap of tandem domain repeats and is given by

$$d_{PQ}^{Dup} = 1 - \exp\left[ - \sum_{i=1}^{N'_P + N'_Q} \frac{|N_i^P - N_i^Q|}{S} \right],$$

$$\text{where } S = \sum_{i=1}^{N'_P + N'_Q} \max(N_i^P, N_i^Q); \tag{6}$$

$N'_P$ and $N'_Q$ are unique domains in proteins P and Q with $N_i^P$ and $N_i^Q$ repeats, respectively.

To compute distances between structures of pairs of ligases, we used AlphaFold2 models (version 4)[68]. We restricted comparisons to contiguous protein segments containing all catalytic domains for each protein to avoid comparing flexible regions of the full-length structures. We computed the TM-score as implemented in US-align[69]. The TM-score between the 3D structures of proteins P and Q is given by,

$$\mathrm{TM\text{-}score}(P, Q) = \max\left[ \frac{1}{L_P} \sum_i^{L_{ali}} \frac{1}{1 + \left(\frac{d_i}{d_0(L_P)}\right)^2} \right], \tag{7}$$

where $L_P$ is the length of protein P, $L_{ali}$ is the number of common residues between aligned proteins P and Q, and $d_0(L_P) = (1.24\sqrt[3]{L_P - 15}) - 1.8$[69]. To account for the inherent

asymmetry in the TM similarity scores due to normalization by reference protein length $L_P$, we computed the structural distance between protein structures P and Q by averaging their TM similarities as

$$d_{PQ}^{Str} = 1 - \frac{TM-score(P,Q) + TM-score(Q,P)}{2}. \tag{8}$$

Functional distances among the protein pairs P and Q were captured using semantic similarities of annotated GO terms corresponding to the three GO ontologies—molecular functions, biological processes, and cellular components—using the package GOGO[70]. The protein to GO term mappings were retrieved (in Feb. 2023) from the Open Biological and Biomedical Ontology Foundry and the Gene Ontology resource[48,71]. For each annotated GO term x, we obtained a directed acyclic graph $DAG_x = (x, T_x, E_x)$ with nodes $T_x$ and edges $E_x$. We defined the semantic contribution, following Wang et al.[72], $S_x(t)$ of a GO term t to the target term x as

$$S_x(t) = \begin{cases} 1 & \text{if } t = x, \\ \max\left(w_e S_x(t') \mid t' \in children(t)\right) & \text{if } t \neq x. \end{cases}$$

Further, the semantic similarity between two GO terms x and y, represented by two graphs $DAG_x$ and $DAG_y$, is defined as

$$Sim_{Wang}(x,y) = \frac{\sum_{t \in T_x \cap T_y} S_x(T) + S_y(t)}{\sum_{t \in T_x} S_x(t) + \sum_{t \in T_y} S_y(t)}.$$

By extension, the semantic similarity between a single GO term x and a set of GO terms $GO_Y = \{y_1, y_2, \cdots, y_k\}$ is defined as the maximum semantic similarity between x and any of the terms in Y:

$$Sim(x, GO_Y) = \max_{1 \leq i \leq k} Sim_{Wang}(x, y_i).$$

Finally, the semantic distance between proteins P and Q, annotated with sets of GO terms $GO_P = \{p_1, p_2, \cdots, p_m\}$ and $GO_Q = \{q_1, q_2, \cdots, q_n\}$, respectively, is calculated as

$$\begin{aligned} d_{PQ}^{Sem} &= 1 - Sim(GO_P, GO_Q) \\ &= 1 - \frac{\sum_{1 \leq i \leq m} Sim(p_i, GO_Q) + \sum_{1 \leq j \leq n} Sim(q_j, GO_Q)}{m+n} \end{aligned} \tag{9}$$

Using Eq. (9), we computed three semantic distances $d_{PQ}^{BP}$, $d_{PQ}^{CC}$, and $d_{PQ}^{MF}$ for the three different GO ontologies.

To compute the subcellular localization distance $d_{PQ}^{ScL}$, each protein's main and auxiliary subcellular locations were mapped from the Human Protein Atlas[73] and used to construct a location vector with weights 1 and 0.3, respectively. We then computed $d_{PQ}^{ScL}$ using the cosine similarity between the location vectors of proteins P and Q as

$$d_{PQ}^{ScL} = 1 - \frac{\mathbf{P} \cdot \mathbf{Q}}{\|\mathbf{P}\| \|\mathbf{Q}\|}. \tag{10}$$

Finally, we computed the tissue ($d_{PQ}^{TE}$) and cell line co-expression ($d_{PQ}^{CIE}$) distances from the tissue and cell line expression profiles of the proteins P and Q. We retrieved expression data from the Human Protein Atlas[73], transcripts per millions of mRNA levels from the 253 human tissues of RNA HPA tissue gene dataset and 1055 cell lines of RNA HPA cell line gene dataset, respectively. Both distances were calculated using Spearman's rank correlation coefficient $r_{S,PQ}$ as

$$d_{PQ}^{TE} = 1 - \frac{1 + r_{S,PQ}^{TE}}{2} \text{ and} \tag{11}$$

$$\begin{aligned} d_{PQ}^{CIE} &= 1 - \frac{1 + r_{S,PQ}^{CIE}}{2}, \\ \text{where } r_{S,PQ} &= \frac{cov(R(P), R(Q))}{\sigma_{R(P)} \sigma_{R(Q)}}. \end{aligned} \tag{12}$$

## Metric optimization, clustering, bootstrapping, and classification

We combined the pairwise gamma ($d_{PQ}^{\gamma}$), Jaccard ($d_{PQ}^{Jac}$), structural ($d_{PQ}^{Str}$), and semantic molecular function ($d_{PQ}^{MF}$) distances to capture all orthogonal information from the four significant hierarchies—sequence, domain architecture, 3D structure, and molecular function—into a single metric spanning the entire molecular scale. We used a weighted-sum model of these four distances, $D_{PQ} = \sum_{i=1}^{4} w_i d_{PQ}^i$, by uniformly sampling the weights as a function of tree cutoff, $h$, a hyperparameter. Optimized weights, $\widehat{w}_i$ were obtained by maximizing the element-centric similarity index[22], which represents the similarity between clusters derived from parsing the emergent dendrogram, at evenly spaced cutoffs, $h \in (0, 1)$ derived from the combined distance and the class-level grouping of E3s into RING, HECT, and RBR classes (partial ground truth). At each cutoff $h$, we sampled ~$10^4$ emergent distance matrices ($\sum_i w_i d_i$), obtained their emergent hierarchical clusters, and computed $S_{EC}$ for each one of them. We chose 100 emergent metrics with the highest $S_{EC}$ for each $h$ and computed the averages and standard deviations of their corresponding weights. The stabilized weights $\widehat{w}_i$ at $h \geq 0.9$ corresponding to the maximum $S_{EC}$ were chosen to construct the optimized distance measure. Dendrograms were computed from hierarchical clustering of individual and combined distance matrices using Ward's minimum variance method[74] as implemented in SciPy. The emergent metric was resampled 500 times by swapping protein labels to compute bootstrap support at each bifurcation node. Unrooted trees with scaled distances were drawn and annotated with domain architectures of individual E3 leaves using iToL[75]. The final tree was parsed at tree cutoff $h = 0.25$ to produce optimal emergent clusters (E3 families). Each family was manually analyzed for shared sequence and domain-architectural features to identify subfamilies and outliers.

## Gene essentiality and CRISPR-Cas9 dropout screens

A pooled multiplexed CRISPR-Cas9 library targeting 822 genes of the human ubiquitin-proteasome system (UPS) was generated according to previously reported methods[76,77]. Each UPS gene was targeted by 2 gRNAs, each expressed from a human 7SK promoter of a dual gRNA-containing lentiviral plasmid, co-expressing puromycin. The second gRNA cassette expressed an AAVS1-targeting gRNA as a neutral control. NGS verified successful cloning, and lentiviral particles were obtained from VectorBuilder.

Library preparation was done using RPE1 cells obtained from ATCC stably expressing Cas9. Cell lines were routinely tested for Mycoplasma contamination. They were transduced with infectious lentiviral particles of the UPS library at 50X coverage and a multiplicity of infection (MOI) of 0.5 to ensure predominantly single integrations per cell. Following infection, cells were subjected to two rounds of puromycin selection (each round for 48 h). Throughout the screen, cells were maintained at a minimum coverage of 50-fold per sgRNA to preserve representation of the library. After ~10 cell doublings ($\approx$ 14 days), cells were harvested and genomic DNA was extracted following previously established protocols[76].

NGS preparation was performed in two steps. First, 900 $\mu$g (50X coverage) of genomic DNA was used in a 50 $\mu$L PCR-1 reaction with primers amplifying both sgRNA library cassettes. PCR-1 had the following composition: 25 $\mu$L of NEBNext® Ultra™ II Q5® Master Mix (NEB,

M0544L), 2 $\mu$L of primer mix (0.4 $\mu$M each, IDT), 2.25 $\mu$g genomic DNA, and H$_2$O to 50 $\mu$L. PCR cycling conditions were as follows: 1 min at 98 °C, followed by 15 s at 98 °C, 20 s at 65 °C, and 60 s at 72 °C for 20 cycles, with a final extension for 60 s at 72 °C using an Eppendorf Mastercycler® X50. PCR-1 products from the same sample were then pooled. For each sample, a second PCR reaction was performed to add P5 and P7 Illumina adaptors, including demultiplexing barcodes. PCR-2 primers also included eight different stagger lengths, all of which were pooled in equimolar amounts. PCR-2 reaction mix was as follows: 25 $\mu$L of NEBNext® Ultra™ II Q5® Master Mix (NEB, M0544L), 1 $\mu$L of P5 primer and 1 $\mu$L P7 primer, including barcodes (0.2 $\mu$M each, SIGMA), 10% input from PCR1 and H$_2$O to 50 $\mu$L. PCR cycling conditions were as follows: 1 min at 98 °C, followed by 15 s at 98 °C, 20 s at 68 °C, and 30 s at 72 °C for 11 cycles, with a final extension for 60 s at 72 °C using an Eppendorf Mastercycler® X50. Barcoded PCR-2 products were then pooled and gel-purified according to the Zymoclean Gel DNA Recovery Kit (Zymo, D4001). NGS Sequencing was performed at Genewiz. Read Counting of the library and both biological replicates of end time points were done with ReCo[78]. For each replicate, we performed log$_2$($CPM + 1$) transformation for each gRNA to obtain normalized read counts.

$$NRC_i = \log_2\left(\frac{gRNA_i \times n \times 100}{\sum_{i=1}^{n} gRNA_i} + 1\right) \qquad (13)$$

Where gRNA$_i$ = RC of gRNA targeting gene $i$; $n$ = total number of guides in the library. We computed log-fold changes (LFCs) as the differences between the NRC$_i$ of both gRNAs. Gene-level LFCs were obtained by averaging the LFC values of guides targeting the same gene. Using Limma[79], we fit a linear model for each gene, considering the different gRNAs as technical replicates and the biological replicates as blocks. We used the `Correlation` function to estimate the intra-block correlation. Then, we calculated $p$ values using empirical Bayes (eBayes) followed by correction for multiple testing using a Benjamini-Hochberg correction. We defined genes with significant contribution to cell fitness if their corresponding $|LFC_i| \geq 1$ and FDR $\leq 0.05$.

### Identifying generic and specific functions of the E3 ligome

GO enrichment analysis for E3 ligases corresponding to individual 13 families was performed using Metascape[80], which implements a hierarchical clustering approach based on $\kappa$-similarity $\geq 0.3$[80]. The resulting networks of GO terms at the biological process, cellular component, and molecular function ontologies were rendered using Cytoscape. Nodes were colored and drawn as pie charts to reflect E3 family contribution (number of proteins) and enrichment. Individual GO terms were considered significantly enriched within a ligase family if the enrichment factor, $C_{obs.}/C_{exp.} \geq 2$, a minimum of 3 proteins corresponding to the family are annotated explicitly with the corresponding GO terms, and a $p$ value $\leq 0.01$. Within each resulting GO cluster, the GO term with the lowest $p$ value was selected as the cluster label for visualization. Heatmaps showing the enriched GO clusters for each family were drawn to highlight the functional specialization of individual E3 families. We ensured high-confidence evidence codes (Experimental, Phylogenetic, and Computational evidence) for enriched GO terms. GO enrichment analysis for essential E3 ligases (background all E3 ligases present in UPS CRISPR dataset) was performed using ShinyGO v0.82[81], employing a Hypergeometric test with Benjamini−Hochberg correction, FDR $\leq 0.05$.

### Cell culture, sample preparation, mass spectrometry data collection, and analysis

HeLa cells were either challenged with (i) global starvation (EBSS treatment triggering nutrient starvation and activating autophagy) or (ii) DNA damage (CPT treatment inhibiting topoisomerase I to induce single-ended DNA double-strand breaks). Cells (ATCC) were cultured in DMEM supplemented with 10% FBS and 100 I.U./mL penicillin-streptomycin in a 5% CO2 atmosphere at 37 °C. Starvation response was induced by incubating cells in EBSS medium (Gibco) for 6h. For triggering DNA damage response, cells were treated with 100 nM Camptothecin for 4h before sample preparation. Cells were regularly tested for Mycoplasma contamination using the Mycoplasma PCR Detection Kit (Sigma). Cells were lysed in Lysis Buffer (2% SDS, 50mM Tris pH 8.5, 10 mM TCEP, 40 mM CAA, supplemented with protease inhibitor cocktail and phosphatase inhibitors), sonicated at 4 °C in a Bioruptor Pico2 (30/30, 10 cycles), and boiled at 95 °C. Proteins were precipitated using methanol-chloroform and digested with 1:50 w/w LysC (Wako Chemicals) and 1:100 w/w trypsin (Promega) overnight at 37 °C. De-salted peptides were dried and resuspended in TMT-labeling buffer (200 mM EPPS pH 8.2, 20% acetonitrile) before being subjected to TMT labeling with a 1:2.5 peptide TMT ratio (w/w) for 1 h at room temperature. The labeling reaction was quenched by the addition of 0.5% hydroxylamine final concentration. Successful TMT labeling was verified by mixing equimolar ratios of peptides and subjecting the mix to single-shot LC-MS/MS analysis. Peptides were fractionated using high-pH liquid chromatography on a micro-flow HPLC (Dionex U3000 RSLC, Thermo Scientific). Pooled fractions were dried in a vacuum concentrator and resuspended in 2% ACN, 0.1% TFA for LC-MS analysis.

Tryptic peptides were analyzed on an Orbitrap Fusion Lumos coupled to an easy nLC 1200 (ThermoFisher Scientific) using a 35 cm long, 75 $\mu$m ID fused-silica column packed in-house with 1.9 $\mu$m C18 particles (Reprosil pur, Dr. Maisch), and kept at 50 °C using an integrated column oven (Sonation). Peptides were eluted from 8% to 28% Buffer B (80% ACN, 0.1% FA) over 75 minutes, followed by a step-wise increase to 90% Buffer B in 21 minutes, which was held for another 9 minutes. A synchronous precursor selection (SPS) multi-notch MS3 method was used[82]. Full scan MS spectra (350-1400 m/z): resolution of 120,000 at m/z 200, maximum injection time of 100 ms, and AGC target value of 4 × 105. MS2 scans (of precursors with charge state between 2-6): maximum injection time of 50 ms, AGC target value of 15000, CID fragmentation with a normalized collision energy (NCE) of 35%. MS3: 10 most intense MS2 fragment ions with an isolation window of 0.7 Th (MS) and 2 m/z (MS2), HCD fragmentation, NCE 50%, resolution of 50000 at m/z 200, scan range of 100-500 m/z, AGC target value of 150000, and a maximum injection time of 86 ms. Repeated sequencing of already acquired precursors was limited by setting a dynamic exclusion of 60 seconds and 7 ppm, and advanced peak determination was deactivated. All spectra were acquired in centroid mode.

MS raw data were analyzed using FragPipe v21.1, with MSFragger v.4.0[83] and Philosopher v.5.1.0[84]. Acquired spectra were searched against the human reference proteome (Taxonomy ID 9606) downloaded from UniProt (07/25/2024; 20,418 entries) with a precursor mass tolerance of 20 ppm and fragment mass tolerance of 20 ppm. Identifications were filtered to obtain false discovery rates (FDR) below 1% for both peptide spectrum matches (minimum peptide length of 7) and proteins using a target-decoy strategy. For all searches, carbamidomethylated cysteine was set as a fixed modification and oxidation of methionine and N-terminal protein acetylation as variable modifications with allowing up to 3 modifications per peptide. Strict trypsin cleavage was set as the protein digestion rule. Label-free quantification was performed using IonQuant v.1.10.27[85]. The differential expression analysis was performed within FragPipe analyst[86] using Limma[79], and derived $p$ values were corrected for multiple comparisons using Benjamini-Hochberg. The mass spectrometry proteomics data have been deposited to the ProteomeXchange Consortium via the PRIDE partner repository with the dataset identifier PXD067015.

### Integrating PPI and ESI datasets

To identify E3 ligases likely functioning in complex mode, we combined data from PDB (https://www.rcsb.org/) and IntAct[87]. Using the refined lists of proteins corresponding to the E3 ligome, E1, E2,

adaptors, receptors, and scaffold proteins (Ubihub and manually curated lists), we retrieved all the PDB structures (as of Jul. 2025) involving E3-adaptors, E3-receptors, and E3-scaffold protein complexes. Additionally, PPIs obtained between E3-adaptor, E3-receptor, and E3-scaffold proteins filtered ("experimentally validated" PPIs, MI:0045 ≥ 0.5) for high confidence enteries. E3s interacting, or in a resolved structure, with at least one receptor, adaptor, or scaffold protein, were re-annotated as complex E3s. To assemble the E3–substrate interaction map, we integrated multiple data sources, including experimentally validated enzyme-substrate interactions (known ESIs) from UbiNet 2.0[57] and UbiBrowser[88], a set of predicted ESIs from UbiBrowser (top 1% of predictions), physically interacting protein pairs (PPIs) from the IntAct database (mapped PPIs), and indirect PPIs involving ligases and potential substrates mediated by adaptor, receptor, or scaffold proteins from IntAct (indirect PPIs).

To obtain a cut-off for filtering PPIs, we first detected a subset of the well-known ESIs (experimentally curated from UbiNet; blue) overlapping with the direct PPIs obtained from IntAct. The PSI-MI confidence scores for these overlapping ESIs ($n = 239$) display a median value of 0.55. This encouraged us to use a threshold value ≥ 0.5 for filtering PPIs. This cutoff strikes a balance between including enough interactions for meaningful analysis while excluding lower-confidence edges that disproportionately contribute noise. Further, major PPI resources (IntAct/IMEx, Reactome, Open Targets) typically use 0.4–0.45 as a medium-confidence baseline. Rounding that to 0.5 provides us with a threshold that sits squarely in the "medium-to-high" confidence range, increasing the proportion of PPIs detected using diverse approaches[89].

Known ESIs and the PPIs dataset were enriched using substrates detected mainly by pull-down experiments, followed by two-hybrid techniques. A map of the ubiquitinated human proteome was obtained by cross-checking the ubiquitination status and mapping ubiquitination sites for each identified substrate from dbPTM[90] and PhosphoSitePlus[91]. All substrates were categorized based on their interactions with E3 ligases: those paired with a single, unique E3 ligase were classified as E3-specific; those associated with multiple E3 ligases from the same family were designated as family-specific; and those linked to two or more E3 ligases from different families were labeled promiscuous.

**Mapping small molecule interaction data**

A unified dataset of E3 handles (corresponding to all publicly documented PROTACs) and E3 binders targeting specific E3 ligases, adaptors, receptors, and scaffold proteins was obtained by combining data from PROTACpedia (https://protacpedia.weizmann.ac.il), and PROTAC-DB 3.0[92] and ChEMBL v34[93]. All small molecules were uniquely identified by their chemical structure, represented using the canonical SMILES format, and mapped to their target proteins and E3 families. Information from ChEMBL v34 was gathered using an SQL query combining compound data, experimental data, and target protein information, and filtered using data from binding assays using p-ChEMBL values.

$$p - ChEMBL = \log_{10}(Activity), \qquad (14)$$

where activity is given by $IC_{50}$, $EC_{50}$, $K_i$, $K_d$, or some measure of potency in molar units. It enables comparison across different bioactivity types. p-ChEMBL ≥ 6.0 is a commonly used threshold in early-stage screening as a baseline for biologically relevant activity ($\leq 1 \times 10^{-6}$ M)[94]. 2048-bit Morgan fingerprint[95] for each small molecule was obtained using RDKit (2048-bit array, radius = 3, http://www.rdkit.org). Dimensionality reduction was performed using PCA, t-SNE, and UMAP using the Python Scikit-learn package (default parameters) and visualized along a 2D subspace, projections showing the highest variance. Further, clustering of small molecules was performed by fast search and

identification of local density peaks[44] using a scaled Euclidean distance measure defined on the corresponding normalized 2D subspaces. Using threshold values for local density ($\rho = 20$) and nearest neighboring peak ($\delta = 0.1$) demonstrated the superiority of UMAP embedding and resulted in 20 small molecule clusters (indexed by $i$, targeting proteins, $j$, with population $n_i = \sum_j n_{ij}$). For each cluster, $i$, representative small molecules were automatically determined from local density peaks, and their distinct set of targeted proteins $j$ from the E3 ligome were also mapped. We then quantified the binding likelihood of individual clusters to target distinct proteins by computing log-transformed propensities, $LP_{ij}$.

$$LP_{ij} = \log_2\left[ \left( \frac{n_{ij}}{\sum_j n_{ij}} \right) \times \left( \frac{\sum_i \sum_j n_{ij}}{\sum_i n_{ij}} \right) \right] \qquad (15)$$

Compound-E3 interaction structures were retrieved from the PDB by converting compound SMILES to InChI keys with RDKit, querying the EBI UniChem API for all small-molecule PDB entries, and merging them with target-protein PDB entries from our dataset, retaining only entries containing both. E3 pocketome data from ELIOT[45] was then mapped to these compound–E3 pairs.

**Reporting summary**

Further information on research design is available in the Nature Portfolio Reporting Summary linked to this article.

## Data availability

All data supporting the findings are provided in the paper and the Supplementary Information File. The mass spectrometry proteomics data have been deposited to the ProteomeXchange Consortium via the PRIDE partner repository with the dataset identifier PXD067015. All data generated, including source data for all figures, and additional Supplementary data files (1–10), are deposited in Zenodo (https://doi.org/10.5281/zenodo.17771730) under a CC BY-NC-SA 4.0 license. A website on "The Human E3 Ligome" to browse the processed data is also hosted on (https://e3-ligome-91adc4.gitlab.io/index.html) under CC BY-NC-SA 4.0 license.

## Code availability

The codes used to develop and analyse "The Human E3 ligome" and perform this study are deposited in the public Zenodo repository (https://doi.org/10.5281/zenodo.17771730) under a CC BY-NC-SA 4.0 license.

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

## Acknowledgements

We thank Stefan Knapp, Gerhard Hummer, Marcel Heinz, and Matthew Shapira, along with all members of the PROXIDRUGs consortium, for their support and constructive discussion. We thank David Krause for system administration and the Center for Supercomputing, Goethe University Frankfurt (GUF), for computing time on the Goethe-HLR cluster. We thank all members of the Centre for Functional Proteomics GUF, in particular, Martin Adrian-Allgood and Bhavesh Parmar. We thank the German Research Foundation, DFG, for funding the LC-MS system (easy nLC1200, Orbitrap Fusion LUMOS) used in this study (FuGG Project-ID: 403765277). This work is primarily funded by PROXIDRUGs, InnoDATA 1.0, and 2.0 projects (03ZU1109KA and 03ZU2109JA), which are part of the "Clusters4Future" initiative funded by the Federal Ministry of Education and Research (BMBF) to R.M.B. We acknowledge financial support from the Innovative Medicines Initiative 2 Joint Undertaking under grant agreement No. 875510 to A.C. and the Deutsche

Forschungsgemeinschaft Project-ID 259130777-SFB1177 on Selective Autophagy to I.D. and R.M.B.

## Author contributions

Conceptualization: R.M.B. Methodology: A.D., A.C., S.V.N., and R.M.B Software: A.D., A.C., S.V.N., and R.M.B. Validation: A.D., A.C., S.V.N., Y.M., B.M., C.T., C.B., R.C., V.J.S, and T.M. Formal analysis: A.D., A.C., S.V.N., and R.M.B. Investigation: A.D., A.C., S.V.N., J.E., Y.M., B.M., C.T., C.B., R.C., V.J.S., T.M., K.H., I.D., M.K., and R.M.B. Resources: I.D., M.K., and R.M.B Data Curation: A.D., A.C., S.V.N., T.M., Y.M., I.D., M.K., and R.M.B Writing–original draft: A.D., A.C., S.V.N., and R.M.B. Writing–review & editing: A.D., A.C., S.V.N., V.J.S., T.M., I.D., M.K., and R.M.B Visualization: A.D., A.C., S.V.N., and R.M.B. Supervision: R.M.B. Funding acquisition: R.M.B.

## Funding

## Competing interests

R.M.B., M.K., K.H., and I.D. are head scientists at the Frankfurt Competence Center for Emerging Therapeutics (FCET), Goethe Center for (high) technology (Go4Tec), Goethe University, Frankfurt am Main, Germany. The remaining authors declare no competing interests. This manuscript reflects the views of the authors, and neither IMI nor the European Union, EFPIA, nor any associated partners are liable for any use that may be made of the information contained herein. I.D. is a founder/shareholder of Vivlion GmbH and a member of its scientific advisory board. I.D. is also a member of the scientific advisory board of the Boehringer Ingelheim Foundation, the expert committee (for international research leader grants) of the Novo Nordisk Foundation, and the advisory board of Cell and Molecular Cell. I.D. was a founder and consultant of Caraway Therapeutics Inc. M.K. is a co-founder, shareholder, and chief officer of Vivlion GmbH.
