## [Transparent Peer Review file · Nature Communications]

Multi-scale classification decodes the complexity of the human E3 ligome

Corresponding Author: Dr Ramachandra Bhaskara

Version 0:

Reviewer comments:

Reviewer #1

(Remarks to the Author)

The present work is highly valuable and has a significant impact, as PROTACs and E3 ligases are currently of great interest in both scientific and pharmaceutical fields. The classification efforts presented in this study are particularly appreciated, as E3 ligases are often challenging to study, poorly characterized, and sometimes misinterpreted. This systematic approach provides much-needed clarity in an area that remains complex, poorly explored but with a great potential.

Suggested minor reviews:

1. In the section "Interaction landscape of the human E3 ligome", better clarify the difference between scaffold proteins, substrate receptors, and adaptors.
2. In the section "Interaction landscape of the human E3 ligome", line 335, explain what is meant by "holo complex structure". The term may be confusing, as it typically refers to a structure bound to a ligand, inhibitor, etc.
3. The description of the filtering criteria applied to PPI and ESI data is detailed, but it would be helpful to specify the reference databases for each criterion, for example:
 - When referring to experimentally validated interactors (MI:0045), clarify whether the PSI-MI score threshold was selected based on standard criteria or if it was an arbitrary choice.
 - For the p-ChEMBL ≥ 6 filter, explain whether this threshold was chosen based on a biological criterion or if it follows specific conventions in pharmaceutical chemistry.
4. In the part on dimensionality reduction with t-SNE, it might be useful to justify the choice of t-SNE over other methods (e.g., PCA, UMAP). Also, specify whether clustering was performed automatically or if categories were assigned manually.
5. The study mentions the use of PDB structures to assess interactions, but the specific PDB structures considered are not explicitly provided (only the Uniprot associated to the E3 ligase). Including a supplementary file listing all PDB structures used in the analyses would improve transparency and reproducibility, allowing others to verify and build upon the findings.
6. The classification of E3 ligases proposed by the authors is innovative and significantly clearer than existing classifications, making it a valuable addition to the field. The analyses performed are also interesting and well-executed. However, the study is not entirely novel, as a 2022 publication introduced a web-based platform called ELIOT, which presents a conceptually similar approach. Given these similarities, it would be appropriate to cite ELIOT and discuss how the current study compares to or builds upon its methodology. Doing so would enhance the contextualization of the work within the existing literature and highlight its unique contributions. <https://doi.org/10.1111/cbdd.14123>

Some comments:

1. The druggability analysis is particularly interesting, as it considers a diverse range of ligands from multiple databases. This comprehensive approach enhances the study's relevance and applicability. However, the comparison relies on a 2D similarity assessment, which may not always be sufficient. While 2D molecular similarity is a valuable starting point, it does not fully capture the potential for structurally dissimilar molecules to exhibit similar 3D interactions. In some cases, compounds with low 2D similarity may still share key binding characteristics when analyzed in three dimensions.

2. The same limitation applies to the clustering approach. The distance matrix used to classify E3 ligases is based on their 2D structures, but it does not account for the fact that different residues can contribute to similar interaction profiles. Additionally, a single protein may adopt multiple conformations, leading to variations in its interaction potential that are not reflected in a purely 2D-based classification. Given the significant role of 3D structural considerations in molecular interactions, this aspect should either be integrated into the analysis or at least acknowledged as a limitation of the current methodology.

Final evaluation:

The work is well written and has a highly valuable impact in the scientific field. The results are well described and accompanied by attractive and well-made images. The methodology is explained quite well, though adding a bit more detail could be beneficial. Only minor revisions are necessary, specifically adding more details and explanation as described in the previously highlighted points.

Overall, the work is very interesting, presenting partially original data regarding the classification of E3 ligases and proposing an innovative and significantly clearer classification compared to existing ones, making it a valuable addition to the field. However, beyond the minor revisions mentioned earlier, two major comments need to be addressed.

First, the presented study is not entirely novel, as a 2022 publication introduced a web-based platform called ELIOT, which shares a conceptually similar aim. While the methodological approach used in this study differs from that of ELIOT, the final product is quite similar. The key distinction is that ELIOT focuses less on classification and instead provides an easily navigable platform where users can directly access and explore information on E3 ligases, binding sites, known interactors, druggability analysis, and clustering analysis. It would be beneficial to discuss this in the manuscript.

Secondly, while the supplementary files related to the dataset used are well-provided, a file regarding the PDBs used is missing. Results are included as supplementary files, but having a more practical way to access and utilize them would be advantageous. For instance, a final, consolidated file in the form of a structured database could allow the scientific community to directly consult and leverage the findings. Without a clear, comprehensive output, the study leans more towards a review that collects and refines data from literature and various databases rather than a research article presenting new results. Providing a tangible and accessible resource would significantly enhance the usability of this important work, ensuring that the broader community can fully benefit from the significant efforts undertaken in this study.

Reviewer #2

(Remarks to the Author)

The study constructs a multi-scale classification system for human E3 ubiquitin ligases by integrating multi-dimensional data, holding important theoretical value in deciphering the complexity of E3s and their functional specialization mechanisms. The proposed metric learning framework offers new insights for enzyme family classification, presenting certain application value to the field. The reviewer suggests the following revisions:

1. The significant value of this paper lies in that the new classification system can predict novel functions for E3 subfamilies. However, the current analyses rely on computational methods such as GO enrichment and structural predictions, lacking wet lab validation. The absence of validation experiments at present results in the mapping between the classification system and actual biological functions remaining at the level of data correlation, failing to form a logically closed loop of experimental confirmation. Regarding the author's claim of "identified over 60 biological processes enriched with E3s corresponding to distinct families," could 1 - 2 key wet lab experiments be added? For example, the statement that "RING5 E3s are enriched in regulating antiviral response, type-I interferon production, regulation of viral entry, and NF- κ B signaling", and the conclusion that "RBR family E3s specialize in K6-linked ubiquitination" are without ubiquitination experiments.
2. The set of E3s studied here relies on the understanding of catalytic domains and is limited by existing data (only covering three types of proteins: RING/HECT/RBR), potentially missing enzymes with non-canonical catalytic features or those dependent on cofactors. This study more closely resembles a refinement of the three major RING/HECT/RBR families rather than an analysis of the entire human E3 ligome as claimed by the authors. By screening only based on known catalytic domains, "986 proteins with low consensus scores and remain unclassified and unannotated, excluding them from the curated E3 ligome," the study may overlook E3s with non-canonical or unknown catalytic mechanisms, making it difficult to cover all E3 types and limiting the comprehensive understanding of the E3 ligome. Authors need to evaluate the completeness of ubiquitin ligase data.
3. Among the 12 distance metrics established by the authors, only four of them were ultimately selected for protein subfamilies classification. What is the rationale behind these selected four distance metrics? Only the related biological features are important for E3 family. When calculating the distance for expression levels, both protein and mRNA expression levels were used—what is the consistency between them?
4. The paper, especially in the abstract, overemphasizes the degradative function of the ubiquitin-proteasome system ("target specific proteins for degradation"). In reality, the non-degradative functions of ubiquitination are also very important, and a revision is suggested.

5. What about the confidence of protein-protein interaction data on the paper's analysis for "Interaction landscape of the human E3 ligome". Is there any bias from the interaction datasets which can affect the analysis results ?

Version 1:

Reviewer comments:

Reviewer #1

(Remarks to the Author)

The work has been updated according to reviewers' requests, and any doubts have been thoroughly addressed. I have nothing further to add and consider it a very valid work, ready for publication.

Reviewer #3

(Remarks to the Author)

I would like to express my sincere gratitude to the authors for their response to the comments. The authors have addressed the questions raised by Reviewer 2 and provided a relatively detailed discussion. Based on the authors' response, particularly regarding Question 2, I wish to further explore this topic. The refined classification of E3 ligases (E3s) has made research on them more standardized and logical. In this context, what practical guiding significance does this paper hold for mass spectrometry-based high-throughput ubiquitination studies? For instance, the multi-omics study on lung squamous cell carcinoma (PMID: 34358469) involves ubiquitination-based omics analysis. Will the more precise classification of E3s, as reported in this study, help uncover aspects that are difficult to explore under the existing analytical framework—such as more specific drug targets that are hard to identify with current analytical approaches?

Version 2:

Reviewer comments:

Reviewer #3

(Remarks to the Author)

My comments have been addressed and I therefore recommend the revised manuscript for publication.

Response to reviewers:

Multi-scale classification decodes the complexity of the human E3 ligome

Arghya Dutta^{1,2,3†}, Alberto Cristiani^{1,2†}, Siddhanta V. Nikte^{1,2†},
Jonathan Eisert^{1,2}, Yves Matthes¹, Borna Markusic^{1,4}, Cosmin Tudose¹,
Chiara Becht¹, Varun Jayeshkumar Shah¹, Thorsten Mosler¹, Koraljka Husnjak¹,
Ivan Dikic^{1,2,4}, Manuel Kaulich¹, Ramachandra M. Bhaskara^{1,2,4*}

¹Institute of Biochemistry II, Faculty of Medicine, Goethe University,
Theodor-Stern-Kai 7, 60590 Frankfurt am Main, Germany.

²Buchmann Institute for Molecular Life Sciences, Goethe University,
Max-von-Laue Strasse 15, 60438 Frankfurt am Main, Germany.

³ Department of Physics, SRM University-AP, Amaravati 522240, Andhra Pradesh, India.

⁴IMPRS on Cellular Biophysics, Max-von-Laue-Str. 3, 60438, Frankfurt am Main, Germany.

***Corresponding author: R.M.B**

E-mail: bhaskara@med.uni-frankfurt.de

[†]These authors contributed equally to this work.

Reviewer 1 (Remarks to the Author)

The present work is highly valuable and has a significant impact, as PROTACs and E3 ligases are currently of great interest in both scientific and pharmaceutical fields. The classification efforts presented in this study are particularly appreciated, as E3 ligases are often challenging to study, poorly characterized, and sometimes misinterpreted. This systematic approach provides much-needed clarity in an area that remains complex, poorly explored but with a great potential.

Response: We thank the reviewer for their positive and encouraging feedback. We are delighted that our systematic classification of E3 ligases and our focus on PROTAC-related mechanisms are recognized as valuable contributions. We appreciate the reviewer’s acknowledgment of the relevance and potential impact of this work.

Suggested minor revisions

1. In the section “Interaction landscape of the human E3 ligome”, better clarify the difference between scaffold proteins, substrate receptors, and adaptors.

Response: We have clarified the definitions of **scaffold**, **adaptor**, and **receptor** subunits (see **Methods**, **Fig. 1b**, and **Supplementary Fig. 1e-g**). We have added the following paragraph to the **Methods** section:

In multisubunit E3 ubiquitin-ligases—most prominently the CRLs—three functionally distinct classes of subunits cooperate to bring an E2~Ub in proximity to a specific substrate. Large, rigid, central scaffold proteins (e.g., the Cullin family, Cul1–Cul5) organize the ligase complex by simultaneously binding the catalytic RING-finger subunit and the docking sites for adaptors or receptors. Adaptor proteins bridge modules that link a scaffold’s N-terminal docking surface to a separate substrate-receptor. They often recognize a conserved motif on the receptor subunit (e.g., F-box, SOCS-box, WD40 proteins). The receptor proteins are specificity determinants that directly recognize and bind degra motifs on substrates. They define which substrates get ubiquitinated (e.g., Skp2, Keap1, VHL). We independently annotate and classify each of the three classes: 151 adaptors, 106 receptors, and 8 scaffold proteins, and use their PPIs to map distinct substrates for multi-subunit E3s.

2. In the section “Interaction landscape of the human E3 ligome”, line 335, explain what is meant by “holo complex structure”. The term may be confusing, as it typically refers to a structure bound to a ligand, inhibitor, etc.

Response: We have now rephrased “holo complex structure” with “multi-subunit complex structure ” for clarity throughout the manuscript.

3. The description of the filtering criteria applied to PPI and ESI data is detailed, but it would be helpful to specify the reference databases for each criterion, for example:

When referring to experimentally validated interactors (MI:0045), clarify whether the PSI-MI score threshold was selected based on standard criteria or if it was an arbitrary choice. For the p-ChEMBL ≥ 6 filter, explain whether this threshold was chosen based on a biological criterion or if it follows specific conventions in pharmaceutical chemistry.

Response: We thank the reviewer for pointing this out. We have now also mentioned the reference databases along with appropriate filtering steps with threshold values (see Methods; **Supplementary Fig. 20**).

To obtain a cut-off for filtering PPIs, we first detected a subset of the well-known ESIs (experimentally curated from UbiNet; blue) overlapping with the PPIs obtained from IntAct (purple, **Supplementary Fig. 20a**). The PSI-MI confidence scores for these overlapping ESIs ($n = 239$) range between 0.25 and 1.0 with a median value of 0.55 (red line in **Supplementary Fig. 20b**). This encouraged us to use a threshold value ≥ 0.5 for filtering PPIs. This cutoff strikes a balance between including enough interactions for meaningful analysis while excluding lower-confidence edges that disproportionately contribute noise. Further, major PPI resources (IntAct/IMEx, Reactome, Open Targets) typically use 0.4–0.45 as a medium-confidence baseline. Rounding that to 0.5 provides us with a threshold that sits squarely in the “medium-to-high” confidence range, increasing the proportion of PPIs detected using diverse approaches (**Supplementary Fig. S20c**) (Porrás et al., 2020).

For the p-ChEMBL value, we first define it and provide the following clarification.

$$\text{p-ChEMBL} = -\log_{10}(\textit{Activity}), \quad (1)$$

where activity is given by IC_{50} , EC_{50} , K_i , K_d , or some measure of potency in molar units. It enables comparison across different bioactivity types. pChEMBL ≥ 6.0 is a commonly used threshold in early-stage screening as a baseline for biologically relevant activity ($\leq 1 \times 10^{-6}$ M) (Smajić et al., 2023), and also a common default value in chemo-informatics tools like RDkit, and other repositories such as PubChem BioAssay.

4. In the part on dimensionality reduction with t-SNE, it might be useful to justify the choice of t-SNE over other methods (e.g., PCA, UMAP). Also, specify whether clustering was performed automatically or if categories were assigned manually.

Response: We thank the reviewer for this comment. We have now redone the dimensionality reduction comparing three different methods: PCA, t-SNE, and UMAP. For our small molecule dataset ($n = 13620$, including E3 handles and E3 binders), we find that UMAP preserves both local neighborhoods and global structure, offers high cluster separability and exhibits both stable and scalable performance (see **Fig. 6b** and **Supplementary Fig. 21c-d**).

We then used the scaled Euclidean distance in the reduced dimensions (2D subspaces: PCA, t-SNE, or UMAP) as a measure to cluster all the compounds by searching and finding local density peaks Rodriguez and Laio (2014). This approach provided more refined cluster definitions (**Fig. 6b**), allowing identification of cluster representatives, and patterns inherent in the dataset, enabling quantification of association between compound structure and their target proteins (**Supplementary Fig. 22**).

5. The study mentions the use of PDB structures to assess interactions, but the specific PDB structures considered are not explicitly provided (only the Uniprot associated with the E3 ligase). Including a supplementary file listing all PDB structures used in the analyses would improve transparency and reproducibility, allowing others to verify and build upon the findings.

Response: We thank the reviewer for pointing this out. We have now updated our mapping to E3 systems and added additional Data files.

`e3ligome_structures.xlsx`: PDB structures associated with the human E3 ligome and allied proteins.

`e3ligome_modes_of_action.xlsx`: The PDB structures of complex E3s.

`e3ligome_sm_clustering.xlsx`: PDBs corresponding to Compound-E3 interactions.

6. The classification of E3 ligases proposed by the authors is innovative and significantly clearer than existing classifications, making it a valuable addition to the field. The analyses performed are also interesting and well-executed. However, the study is not entirely novel, as a 2022 publication introduced a web-based platform called ELIOT, which presents a conceptually similar approach. Given these similarities, it would be appropriate to cite ELIOT and discuss how the current study compares to or builds upon its methodology. Doing so would enhance the contextualization of the work within the existing literature and highlight its unique contributions (Palomba et al., 2022).

Response: We thank the reviewer for this insightful comment. We are pleased that these aspects of our work are recognized as valuable additions to the field.

We now explicitly cite ELIOT: a valuable resource of E3 pocketome Palomba et al. (2022) and discuss the overlap and complementary aspects of our study, within the purview of analyzing E3 ligase structures and their druggable pockets for E3 handle design (see **Results**, section 6, paragraph 3, **Fig. 6**).

We clarify that our primary goal in the current manuscript is the **classification** of the *human E3 ligome* and how this offers a unique perspective on functional segregation, enables identification of substrates via mapping protein-interaction landscape, and provides a druggability map. Only our last section on “Druggability map of the human E3 ligome” overlaps partially with ELIOT (**Fig. 6**). Here, we prioritize mapping of small-molecule E3 handles and E3 binders onto the new refined classification, highlighting (1) similarity patterns of current E3 binders using clustering in UMAP space, (2) potential avenues for PROTAC re-purposing, and (3) potential identification of new lead compounds for handle design.

A search against the PDB identified structures for 231 compound-E3 pairs. Extending this to ELIOT pocketome data resulted in 74 overlapping pairs (out of the 209 pairs present in ELIOT; **Fig. 6e**). This information proved useful to learn pocket characteristics of potential new E3 binders, which could be developed into lead compounds (**Fig. 6g**). We found that pocket similarity data and PROTAC scores of individual pockets complemented our hit identification process for lead development from initial

E3 binders. ELIOT data provides a clear structural context for drugging structurally resolved E3 pockets.

Some comments (Reviewer 1)

1. The druggability analysis is particularly interesting, as it considers a diverse range of ligands from multiple databases. This comprehensive approach enhances the study’s relevance and applicability. However, the comparison relies on a 2D similarity assessment, which may not always be sufficient. While 2D molecular similarity is a valuable starting point, it does not fully capture the potential for structurally dissimilar molecules to exhibit similar 3D interactions. In some cases, compounds with low 2D similarity may still share key binding characteristics when analyzed in three dimensions.

Response: We thank the reviewer for highlighting the limitations of relying solely on 2D similarity assessments in the context of our E3 druggability map (**Fig. 6b**). 2D fingerprints may overlook critical aspects of molecular behavior that become apparent only in 3D, such as shape complementarity, distinguishing stereoisomers, and conformational flexibility.

Morgan (circular) fingerprints paired with the clustering remain among the most popular tools for rapid similarity searching in chemo-informatics. 2D Morgan fingerprints are topological descriptors that encode the presence of atom-centered substructures. They can be computed with relative ease, speed (milliseconds for 1000 compounds), and flexibility (alter radius and bit-length). They are easy to interpret (each bit corresponds to a specific local substructure) and consistently rank among the top-performing fingerprints in QSAR/QSPR modeling, scaffold hopping, and target prediction benchmarks (Capecchi et al., 2020). Furthermore, using retrospective benchmarks, e.g., the Directory of useful Decoys (DUD dataset; <http://dud.docking.org/>), have shown that 2D fingerprints often outperform 3D-shape methods in early enrichment metrics, because many active compounds share core substructures that drive binding (Hu et al., 2012).

We now include a more explicit discussion on the trade-offs of using 2D descriptors. Specifically, we now write (see **Discussion**):

Although 2D-based UMAP visualizations offer a computationally efficient overview of chemical diversity and aid in hypothesis generation, they may fail to capture activity cliffs or bio-isosteric relationships. Complementing the druggability map generated here through physics-based approaches, including 3D conformational sampling and HTP docking screens of ligands against the druggable E3 pocketome (Palomba et al., 2022) will propel the rational design of new E3 handles.

However, extensive 3D conformational sampling and molecular docking of potential new E3 binders are beyond the current scope of the manuscript, which focuses on the **classification problem of human E3 ligases**. Finally, the mapping of small

molecules into well-defined clusters obtained here, aids in identifying specificity features of potential E3 binders to an expanded set of E3 ligases, and must be viewed as a critical hypotheses generation step in (1) establishing the current coverage of the E3 ligome, and (2) bridge gaps in druggability space for the design of new E3 handles.

2. The same limitation applies to the clustering approach. The distance matrix used to classify E3 ligases is based on their 2D structures, but it does not account for the fact that different residues can contribute to similar interaction profiles. Additionally, a single protein may adopt multiple conformations, leading to variations in its interaction potential that are not reflected in a purely 2D-based classification. Given the significant role of 3D structural considerations in molecular interactions, this aspect should either be integrated into the analysis or at least acknowledged as a limitation of the current methodology.

Response: In classifying E3 ligases, we initially used a total of 12 different distance measures from different granular layers (see **Methods**, **Eq. 2–12**). In the final metric learning, we used 4 individual distance measures, explicitly including 3D structures of E3 ligases, d_{PQ}^{Str} (**Fig. 2**), spanning all features from the molecular scale. We used TM-scores (**Eq. 8**) to quantify the structural similarity of E3 ligases. We limit our computations to the catalytically active structured domains and exclude terminal disordered regions (regions with low pLDDT).

TM-scores accurately capture the trends in similarities, given the fluctuations around the X-ray models. Moreover, recent deep learning methods predict TM-scores from protein sequences to establish remote homologues Hamamsy et al. (2023). In fact our final optimization and metric learning exercise, resulted in the highest weight ($\hat{w}_{\text{Str}} = 0.70$) for the 3D structural distance (**Fig 2f** and **Eq. 1**), reaffirming the importance of 3D structure, and their preserved patterns in contributing to family and sub-family organization. Therefore, our distance measurements accurately consider and compare 3D structures of E3 ligases (TM-score based) and are robust to the small internal dynamics of the catalytic domains, if any, during the functional cycle. We emphasize this point more explicitly in the revised manuscript (**Results** section on Metric learning, and **Discussion**).

Functional dynamics is not always captured by static structures (from either PDB or AlphaFold models) and requires detailed characterization of functional conformational changes using explicit molecular dynamics simulations (Wang et al., 2024; Dixon et al., 2022; Zheng et al., 2022; Liwocha et al., 2020; González et al., 2023), beyond the scope of the current manuscript.

Final evaluation (Reviewer 1)

The work is well written and has a highly valuable impact in the scientific field. The results are well described and accompanied by attractive and well-made images. The methodology is explained quite well, though adding a bit more detail could be beneficial. Only minor revisions are necessary, specifically adding more details and explanation as described in the previously highlighted points. Overall, the work is very interesting, presenting partially

original data regarding the classification of E3 ligases and proposing an innovative and significantly clearer classification compared to existing ones, making it a valuable addition to the field. However, beyond the minor revisions mentioned earlier, two major comments need to be addressed.

Response: We thank the reviewer for overall positive feedback on the significance, clarity, and presentation of our work. We have incorporated additional methodological details as suggested and addressed both the major and minor comments to further strengthen the manuscript.

- First, the presented study is not entirely novel, as a 2022 publication introduced a web-based platform called ELIOT, which shares a conceptually similar aim. While the methodological approach used in this study differs from that of ELIOT, the final product is quite similar. The key distinction is that ELIOT focuses less on classification and instead provides an easily navigable platform where users can directly access and explore information on E3 ligases, binding sites, known interactors, druggability analysis, and clustering analysis. It would be beneficial to discuss this in the manuscript.

Response: In the revised version, we now explicitly map ELIOT data onto our compound-E3 interaction pairs, highlighting how ELIOT data complements our drug re-purposing and lead development strategy by providing a clear structural context (**Fig. 6**). We further note that extending ELIOT analysis for a larger dataset of compound-E3 pairs using predicted/docked compound-E3 interaction models from high-throughput in silico screens will reveal a much richer landscape of uncharted pocketome.

- Secondly, while the supplementary files related to the dataset used are well-provided, a file regarding the PDBs used is missing. Results are included as supplementary files, but having a more practical way to access and utilize them would be advantageous. For instance, a final, consolidated file in the form of a structured database could allow the scientific community to directly consult and leverage the findings. Without a clear, comprehensive output, the study leans more towards a review that collects and refines data from literature and various databases rather than a research article presenting new results. Providing a tangible and accessible resource would significantly enhance the usability of this important work, ensuring that the broader community can fully benefit from the significant efforts undertaken in this study.

Response: We now provide a dedicated repository (downloadable https://gitlab.com/ccblab/e3_ligome) and a web-platform (<https://e3-ligome-91adc4.gitlab.io/index.html>) to access “**The human E3 ligome**”. This website provides information on E3 ligome, data collection, filtering process, curation, and final curated data, along with the classification of the E3 ligome into distinct families and sub-families in a downloadable format. Further, for each family, we provide summaries of our findings, along with enriched functions and substrates, and small-molecule interactions. For each E3 catalytic and non-catalytic component protein, we provide database confidence scores, basic information on classification, and hyperlinks (UniProt, GeneCards,

InterPro, AlphaFold, PDBs, GO annotations, IntAct, and ChEMBL) to the original raw data used in the *Human E3 ligome* project.

Reviewer 2 (Remarks to the Author)

The study constructs a multi-scale classification system for human E3 ubiquitin ligases by integrating multi-dimensional data, holding important theoretical value in deciphering the complexity of E3s and their functional specialization mechanisms. The proposed metric learning framework offers new insights for enzyme family classification, presenting certain application value to the field. The reviewer suggests the following revisions:

Response: We thank the reviewer for recognizing the value of our multi-scale classification and the potential application of our metric learning framework. We reinforce the focus on how this classification reflects the complexity and functional specialization of E3 ligases by addressing all questions raised.

1. The significant value of this paper lies in that the new classification system can predict novel functions for E3 subfamilies. However, the current analyses rely on computational methods such as GO enrichment and structural predictions, lacking wet lab validation. The absence of validation experiments at present results in the mapping between the classification system and actual biological functions remaining at the level of data correlation, failing to form a logically closed loop of experimental confirmation. Regarding the author’s claim of “identified over 60 biological processes enriched with E3s corresponding to distinct families,” could 1–2 key wet lab experiments be added? For example, the statement that “RING5 E3s are enriched in regulating antiviral response, type-I interferon production, regulation of viral entry, and NF- κ B signaling”, and the conclusion that “RBR family E3s specialize in K6-linked ubiquitination” are without ubiquitination experiments.

Response:

We thank the reviewer for this valuable comment. We have now revised the entire section on “Functional segregation of the human E3 ligome”, and added new experimental data.

(1) We performed CRISPR-cas9 dropout screens for UPS genes, identifying essential catalytic and non-catalytic E3 components for cell viability (see **Fig. 4a-c**). We also show that these findings correlated with the depMap data (**Supplementary Fig. 18a**).

(2) We revisited our functional cluster enrichment analysis (family-specific functions of the E3 ligome) and limited our predictions using only high-confidence GO annotations (see **Methods; Fig. 4d-e, Supplementary Figs. 18-19**). Further, we map explicit previous research (PMIDs) consistent with each functional segregation pattern observed. Predictions using low-confidence annotations are also provided and serve as new hypotheses for future testing (**Supplementary Figs. 19**).

(3) We discuss 2 examples of family-specific functions corresponding to the RBR and RING5 family in detail, providing mechanistic insights (**Fig. 4f**).

(4) We measured starvation and DNA damage response of HeLa cells and identified overlapping E3 ligases with significant changes in proteomics experiments consistent

with our predictions. We outline the mechanistic basis for these E3s in activating autophagic response and DNA damage repair processes (non-degradative functions of the E3 ligome; **Fig. 4g-h**).

2. The set of E3s studied here relies on the understanding of catalytic domains and is limited by existing data (only covering three types of proteins: RING/HECT/RBR), potentially missing enzymes with non-canonical catalytic features or those dependent on cofactors. This study more closely resembles a refinement of the three major RING/HECT/RBR families rather than an analysis of the entire human E3 ligome as claimed by the authors. By screening only based on known catalytic domains, “986 proteins with low consensus scores remain unclassified and unannotated, excluding them from the curated E3 ligome,” the study may overlook E3s with non-canonical or unknown catalytic mechanisms, making it difficult to cover all E3 types and limiting the comprehensive understanding of the E3 ligome. Authors need to evaluate the completeness of ubiquitin ligase data.

Response: We thank the reviewer for highlighting these important points.

(1) We now provide detailed definitions of catalytic and non-catalytic components of the E3 ligome (adaptors, receptors, scaffold proteins, and E2 proteins $n = 151 + 106 + 8 + 33 = 298$) and explain the exclusion process (see edited **Fig. 1b-c**, **Supplementary Fig. 1**, and **Methods**). We now exclude only the unclassified proteins and those lacking clear domain annotations, $n = 514 + 174 = 688$, thus accounting for all proteins in the unified datasets. Further, the confidence scores provide an estimate of our current knowledge on each protein.

(2) We now add a dedicated section on the “Completeness of our E3 ligome” (see **Supplementary Text 1**) covering detailed analysis on the errors of omission (missed 5 E3s, True negatives) and commission (included 6 non-E3s, False positives). We discuss the scope and limitations of our curation approach, covering pseudo-E3 ligases, E3s with untested and minimal activity, and other non-catalytic components (E2s, receptors, adaptors, and scaffold proteins).

(3) Capturing atypical or non-canonical ligase mechanisms computationally remains a challenge. Gathering experimental proof for atypical E3 mechanisms is already quite hard and an active area of research. However, in addition to canonical RING, HECT, and RBR mechanisms, our E3 ligome captures several known atypical E3 ligases (see **Supplementary Text 2** for a detailed account). Their inclusion in our E3 ligome is explained by the presence of evolutionarily related RING-, HECT-, or RBR-like domains with atypical mechanisms. The heterogeneous HECT2 family contains an outlier RING-domain containing sub-family, which is populated with atypical E3 catalytic mechanisms (e.g., MYCBP2, RNF213, etc.). We have now highlighted interesting features of our classification in the revised version.

We acknowledge that ongoing research may further expand or refine the human E3 ligome, just as in the case of the human Kinome project. Our current version of the E3 ligome provides a robust and objective foundation for functional analysis, while recognizing existing knowledge gaps. We host this data with version control on our GitLab repository to be updated as more information emerges.

3. Among the 12 distance metrics established by the authors, only four of them were ultimately selected for protein subfamilies classification. What is the rationale behind these selected four distance metrics? Only the related biological features are important for the E3 family. When calculating the distance for expression levels, both protein and mRNA expression levels were used—what is the consistency between them?

Response: We selected only 4/12 distance measures as they span fundamental granularity layers corresponding to the molecular scale (**Fig. 2b**). These four distances capture intrinsic molecular attributes and their relationships. Sequence-, domain composition-, 3D structure-, and GO molecular function-based distances provide accurate information on conserved residues/motifs, preserved catalytic and non-catalytic domains, aligned spatial organization, and comparable enzymatic activity, respectively. These attributes maximize clustering of mechanistic and functionally similar E3s. Furthermore, they reflect E3 evolution, capturing gene/domain duplication, diversification, and rearrangement events, for instance, in co-clustering of paralogs with conserved patterns within the E3 ligome.

By contrast, the other distances based on GO biological processes, cellular components, and sub-cellular localization, cell and tissue expression profiles describe more systems-level organization. These distances have broad and noisy distributions, and are non-specific to known E3 classes, limiting their relevance to classify related E3 mechanisms and functions. Including such noisy features would dilute the classification.

We find that for E3 ligases, the protein abundance (pTPM (protein) from PaxDb) and expression (pTPM (mRNA) from Human Protein Atlas) of HeLa cells are positively correlated (Pearson $r = 0.56$, **Supplementary Fig. 23b**). Further, this data is only available for a few cell and tissue types, limiting its applicability for metric learning.

Further, the computing time and the optimization complexity limit the total number of distance measures for the linear combination metric learning approach here. Current brute-force sampling of all of 10^5 metrics would increase to 10^{13} metric evaluations, making it costly. Although more conservative sampling procedures could reduce the number of metric evaluations and computational cost, their efficiency in exploring informative parameter space remains untested.

For these reasons, we focused on the four most informative distances. However, we include all twelve in the paper for completeness. The other distance measures provide systems-level information on E3 systems more relevant for understanding E3 biology and developing an efficient drugging strategy (see **Supplementary Fig. 23**). More specifically, they provide complementary information in the form of unique projection maps for E3s onto sub-cellular compartments to target distinct substrate proteins, Cell-line expression for new assay development, and tissue-specificity relevant to human physiology.

4. The paper, especially in the abstract, overemphasizes the degradative function of the ubiquitin-proteasome system (“target specific proteins for degradation”). In reality, the non-degradative functions of ubiquitination are also very important, and a revision is suggested.

Response: We thank the reviewer for this comment. We have now added a separate section discussing the non-degradative functions of ubiquitination (see **Supplementary Text 16**). In addition, we have revised the abstract, main text, and discussion to highlight the importance of these non-degradative roles.

5. What about the confidence of protein-protein interaction data on the paper’s analysis for “Interaction landscape of the human E3 ligome”? Is there any bias from the interaction datasets that can affect the analysis results?

Response: The confidence of the protein-protein interaction (PPI) data in the analysis of the “Interaction landscape of the human E3 ligome” largely depends on the stringency of data curation and the filtering process applied to minimize false positives.

To begin with, only experimental interactions (MI:0045) are considered, as annotated in IntAct. This first step filters interactions based on inference, predictions, and unspecified methods, which could otherwise introduce systematic bias. To obtain a cut-off for further filtering of these interactions, we first detected a subset of the well-known ESIs (experimentally curated from UbiNet; blue) overlapping with the direct PPIs obtained from IntAct (purple, **Supplementary Fig. 20a**). The PSI-MI confidence scores for these overlapping ESIs ($n = 239$) range between 0.25 and 1.0 (**Supplementary Fig. 20b**) with a median value of 0.55 (red line in **Supplementary Fig. 20b**). This encouraged us to use a threshold value ≥ 0.5 for filtering PPIs. This cutoff strikes a balance between including enough interactions for meaningful analysis while excluding lower-confidence edges that disproportionately contribute noise. Further, major PPI resources (IntAct/IMEx, Reactome, Open Targets) typically use 0.4–0.45 as a medium-confidence baseline. Rounding that to 0.5 provides us with a threshold that sits squarely in the “medium-to-high” confidence range, increasing the proportion of PPIs detected using diverse approaches (**Supplementary Fig. 20c**) (Porrás et al., 2020).

Further, for known ESIs (UbiNet), the majority of interactions were detected via anti-tag coIP, anti-bait coIP, and pulldown experiments, with only a small number derived from other methods (see **Supplementary Fig. 20c**, top). In contrast, the PPIDirect dataset is dominated by Y2H-based approaches, including standard two-hybrid, validated two-hybrid, and prey-pooling assays, which are designed to identify direct binary interactions (see **Supplementary Fig. 20c**, bottom). Y2H-derived data is particularly valuable because it is more sensitive to transient interactions, such as those between E3 ligases and their potential substrates. Alternatively, CoIP methods might be useful to detect stable multi-subunit complexes containing E3s. This emphasizes the complementary nature of the datasets and the importance of integrating high-confidence Y2H-derived PPIs with CoIP-based PPIs to achieve a more comprehensive and accurate E3-substrate interaction map.

Despite these precautions, PPI datasets could exhibit methodological and literature biases, which are important to consider for E3-substrate studies. For example, interactions from high-throughput experiments (e.g., AP-MS) often over-represent stable complexes, while transient or weak interactions are more relevant to E3-substrate pairs.

There is also a bias towards well-studied proteins, as interactions involving highly characterized ligases or substrates are more likely to be reported in the literature and curated into databases.

References

- P. Porras, E. Barrera, A. Bridge, N. del Toro, G. Cesareni, M. Duesbury, H. Hermjakob, M. Iannuccelli, I. Jurisica, M. Kotlyar, et al., *Nature Communications* **11** (2020), ISSN 2041-1723, URL <http://dx.doi.org/10.1038/s41467-020-19942-z>.
- A. Smajić, I. Rami, S. Sosnin, and G. F. Ecker, *Chemical Research in Toxicology* **36**, 1300–1312 (2023), ISSN 1520-5010, URL <http://dx.doi.org/10.1021/acs.chemrestox.3c00042>.
- A. Rodriguez and A. Laio, *Science* **344**, 1492–1496 (2014), ISSN 1095-9203, URL <http://dx.doi.org/10.1126/science.1242072>.
- T. Palomba, M. Baroni, S. Cross, G. Cruciani, and L. Siragusa, *Chemical Biology & Drug Design* **101**, 69–86 (2022), ISSN 1747-0285, URL <http://dx.doi.org/10.1111/cbdd.14123>.
- A. Capecchi, D. Probst, and J.-L. Reymond, *Journal of Cheminformatics* **12** (2020), ISSN 1758-2946, URL <http://dx.doi.org/10.1186/s13321-020-00445-4>.
- G. Hu, G. Kuang, W. Xiao, W. Li, G. Liu, and Y. Tang, *Journal of Chemical Information and Modeling* **52**, 1103–1113 (2012), ISSN 1549-960X, URL <http://dx.doi.org/10.1021/ci300030u>.
- T. Hamamsy, J. T. Morton, R. Blackwell, D. Berenberg, N. Carriero, V. Gligorijevic, C. E. M. Strauss, J. K. Leman, K. Cho, and R. Bonneau, *Nature Biotechnology* **42**, 975–985 (2023), ISSN 1546-1696, URL <http://dx.doi.org/10.1038/s41587-023-01917-2>.
- Z. Wang, F. Fan, Z. Li, F. Ye, Q. Wang, R. Gao, J. Qiu, Y. Lv, M. Lin, W. Xu, et al., *Nature Communications* **15** (2024), ISSN 2041-1723, URL <http://dx.doi.org/10.1038/s41467-024-47586-w>.
- T. Dixon, D. MacPherson, B. Mostofian, T. Dauzhenka, S. Lotz, D. McGee, S. Shechter, U. R. Shrestha, R. Wiewiora, Z. A. McDargh, et al., *Nature Communications* **13** (2022), ISSN 2041-1723, URL <http://dx.doi.org/10.1038/s41467-022-33575-4>.
- S. Zheng, Y. Tan, Z. Wang, C. Li, Z. Zhang, X. Sang, H. Chen, and Y. Yang, *Nature Machine Intelligence* **4**, 739–748 (2022), ISSN 2522-5839, URL <http://dx.doi.org/10.1038/s42256-022-00527-y>.
- J. Liwocha, D. T. Krist, G. J. van der Heden van Noort, F. M. Hansen, V. H. Truong, O. Karayel, N. Purser, D. Houston, N. Burton, M. J. Bostock, et al., *Nature Chemical Biology* **17**, 272–279 (2020), ISSN 1552-4469, URL <http://dx.doi.org/10.1038/s41589-020-00696-0>.

A. González, A. Covarrubias-Pinto, R. M. Bhaskara, M. Glogger, S. K. Kuncha, A. Xavier, E. Seemann, M. Misra, M. E. Hoffmann, B. Bräuning, et al., *Nature* **618**, 394–401 (2023), ISSN 1476-4687, URL <http://dx.doi.org/10.1038/s41586-023-06089-2>.

Response to reviewers:

Multi-scale classification decodes the complexity of the human E3 ligome

Arghya Dutta^{1,2,3†}, Alberto Cristiani^{1,2†}, Siddhanta V. Nikte^{1,2†},
Jonathan Eisert^{1,2}, Yves Matthes¹, Borna Markusic^{1,4}, Cosmin Tudose¹,
Chiara Becht¹, Varun Jayeshkumar Shah¹, Thorsten Mosler¹, Koraljka Husnjak¹,
Ivan Dikic^{1,2,4}, Manuel Kaulich¹, Ramachandra M. Bhaskara^{1,2,4*}

¹Institute of Biochemistry II, Faculty of Medicine, Goethe University,
Theodor-Stern-Kai 7, 60590 Frankfurt am Main, Germany.

²Buchmann Institute for Molecular Life Sciences, Goethe University,
Max-von-Laue Strasse 15, 60438 Frankfurt am Main, Germany.

³ Department of Physics, SRM University-AP, Amaravati 522240, Andhra Pradesh, India.

⁴IMPRS on Cellular Biophysics, Max-von-Laue-Str. 3, 60438, Frankfurt am Main, Germany.

***Corresponding author: R.M.B**

E-mail: bhaskara@med.uni-frankfurt.de

[†]These authors contributed equally to this work.

Reviewer comments on the Revised Manuscript

Reviewer 1 (Remarks to the Author)

The work has been updated according to reviewers' requests, and any doubts have been thoroughly addressed. I have nothing further to add and consider it a very valid work, ready for publication.

Response: We thank the Reviewer #1 for their suggestions in improving the manuscript.

Reviewer 3 (Remarks to the Author)

I would like to express my sincere gratitude to the authors for their response to the comments. The authors have addressed the questions raised by Reviewer 2 and provided a relatively detailed discussion.

Response: We greatly appreciate Reviewer #3's kind remark and are pleased that our revisions and detailed responses have satisfactorily addressed all previous questions. We thank the reviewer for recognizing our efforts to improve the manuscript.

Based on the authors' response, particularly regarding Question 2, I wish to further explore this topic. The refined classification of E3 ligases (E3s) has made research on them more standardized and logical. In this context, what practical guiding significance does this paper hold for mass spectrometry-based high-throughput ubiquitination studies? For instance, the multi-omics study on lung squamous cell carcinoma (PMID: 34358469) involves ubiquitination-based omics analysis. Will the more precise classification of E3s, as reported in this study, help uncover aspects that are difficult to explore under the existing analytical framework—such as more specific drug targets that are hard to identify with current analytical approaches?

Response: To address this additional query, we added **Supplementary Text 17** (cited in **Results** and **Discussion** sections), outlining practical and actionable guidelines on how the proposed human E3 ligome can be harnessed.

In **Supplementary Text 17**, we describe how our classification framework enables systematic annotation of large-scale omics datasets by integrating diverse HTP screening data—from proteomics to genetic and small-molecule screening studies to uncover previously uncharacterized E3s, substrates, and chemical modulators. The E3 ligome structure provides a framework for validating family- and subfamily-specific functions, mapping degrons, and identifying CRISPR-based genetic dependencies. We also highlight how mechanistic screens using catalytic mutants or activity-based probes can assess functional convergence within E3 families. Finally, integrating the E3 ligome data obtained here with spatial transcriptomics, phenotypic profiling and chemical proteomics can reveal context-specific E3s and inform rational PROTAC design, underscoring the broad analytical and experimental utility of our framework. In summary, we envision that the human E3 ligome classification presented here will catalyze E3 biology much like the Human Kinome project (Manning et al., 2002) transformed kinase research.

References

- P. Porras, E. Barrera, A. Bridge, N. del Toro, G. Cesareni, M. Duesbury, H. Hermjakob, M. Iannuccelli, I. Jurisica, M. Kotlyar, et al., *Nature Communications* **11** (2020), ISSN 2041-1723, URL <http://dx.doi.org/10.1038/s41467-020-19942-z>.
- A. Smajić, I. Rami, S. Sosnin, and G. F. Ecker, *Chemical Research in Toxicology* **36**, 1300–1312 (2023), ISSN 1520-5010, URL <http://dx.doi.org/10.1021/acs.chemrestox.3c00042>.
- A. Rodriguez and A. Laio, *Science* **344**, 1492–1496 (2014), ISSN 1095-9203, URL <http://dx.doi.org/10.1126/science.1242072>.
- T. Palomba, M. Baroni, S. Cross, G. Cruciani, and L. Siragusa, *Chemical Biology & Drug Design* **101**, 69–86 (2022), ISSN 1747-0285, URL <http://dx.doi.org/10.1111/cbdd.14123>.
- A. Capecchi, D. Probst, and J.-L. Reymond, *Journal of Cheminformatics* **12** (2020), ISSN 1758-2946, URL <http://dx.doi.org/10.1186/s13321-020-00445-4>.
- G. Hu, G. Kuang, W. Xiao, W. Li, G. Liu, and Y. Tang, *Journal of Chemical Information and Modeling* **52**, 1103–1113 (2012), ISSN 1549-960X, URL <http://dx.doi.org/10.1021/ci300030u>.
- T. Hamamsy, J. T. Morton, R. Blackwell, D. Berenberg, N. Carriero, V. Gligorijevic, C. E. M. Strauss, J. K. Leman, K. Cho, and R. Bonneau, *Nature Biotechnology* **42**, 975–985 (2023), ISSN 1546-1696, URL <http://dx.doi.org/10.1038/s41587-023-01917-2>.

- Z. Wang, F. Fan, Z. Li, F. Ye, Q. Wang, R. Gao, J. Qiu, Y. Lv, M. Lin, W. Xu, et al., *Nature Communications* **15** (2024), ISSN 2041-1723, URL <http://dx.doi.org/10.1038/s41467-024-47586-w>.
- T. Dixon, D. MacPherson, B. Mostofian, T. Dauzhenka, S. Lotz, D. McGee, S. Shechter, U. R. Shrestha, R. Wiewiora, Z. A. McDargh, et al., *Nature Communications* **13** (2022), ISSN 2041-1723, URL <http://dx.doi.org/10.1038/s41467-022-33575-4>.
- S. Zheng, Y. Tan, Z. Wang, C. Li, Z. Zhang, X. Sang, H. Chen, and Y. Yang, *Nature Machine Intelligence* **4**, 739–748 (2022), ISSN 2522-5839, URL <http://dx.doi.org/10.1038/s42256-022-00527-y>.
- J. Liwocha, D. T. Krist, G. J. van der Heden van Noort, F. M. Hansen, V. H. Truong, O. Karayel, N. Purser, D. Houston, N. Burton, M. J. Bostock, et al., *Nature Chemical Biology* **17**, 272–279 (2020), ISSN 1552-4469, URL <http://dx.doi.org/10.1038/s41589-020-00696-0>.
- A. González, A. Covarrubias-Pinto, R. M. Bhaskara, M. Glogger, S. K. Kuncha, A. Xavier, E. Seemann, M. Misra, M. E. Hoffmann, B. Bräuning, et al., *Nature* **618**, 394–401 (2023), ISSN 1476-4687, URL <http://dx.doi.org/10.1038/s41586-023-06089-2>.
- G. Manning, D. B. Whyte, R. Martinez, T. Hunter, and S. Sudarsanam, *Science* **298**, 1912–1934 (2002), ISSN 1095-9203, URL <http://dx.doi.org/10.1126/science.1075762>.